# Metabolic reprogramming and membrane glycan remodeling as potential drivers of zebrafish heart regeneration

Renza Spelat [1,2,10], Federico Ferro [1,3,10], Paolo Contessotto[1,4,10], Amal Aljaabary [1], Sergio Martin-Saldaña [1], Chunsheng Jin [5], Niclas G. Karlsson [5], Maura Grealy[6], Markus M. Hilscher [7], Fulvio Magni[8], Clizia Chinello[8], Michelle Kilcoyne [1,9] & Abhay Pandit [1✉]

The ability of the zebrafish heart to regenerate following injury makes it a valuable model to deduce why this capability in mammals is limited to early neonatal stages. Although metabolic reprogramming and glycosylation remodeling have emerged as key aspects in many biological processes, how they may trigger a cardiac regenerative response in zebrafish is still a crucial question. Here, by using an up-to-date panel of transcriptomic, proteomic and glycomic approaches, we identify a metabolic switch from mitochondrial oxidative phosphorylation to glycolysis associated with membrane glycosylation remodeling during heart regeneration. Importantly, we establish the N- and O-linked glycan structural repertoire of the regenerating zebrafish heart, and link alterations in both sialylation and high mannose structures across the phases of regeneration. Our results show that metabolic reprogramming and glycan structural remodeling are potential drivers of tissue regeneration after cardiac injury, providing the biological rationale to develop novel therapeutics to elicit heart regeneration in mammals.

[1] CÚRAM, SFI Research Centre for Medical Devices, University of Galway, Galway, Ireland. [2] Neurobiology Sector, International School for Advanced Studies (SISSA), Trieste, Italy. [3] Department of Medical Surgery and Health Science, University of Trieste, Trieste, Italy. [4] Department of Molecular Medicine, University of Padova, Padova, Italy. [5] Department of Medical Biochemistry, Institute of Biomedicine, Sahlgrenska Academy, University of Gothenburg, Gothenburg, Sweden. [6] Pharmacology and Therapeutics, School of Medicine, University of Galway, Galway, Ireland. [7] Science for Life Laboratory, Department of Biochemistry and Biophysics, Stockholm University, Stockholm, Sweden. [8] Clinical Proteomics and Metabolomics Unit, School of Medicine and Surgery, University of Milano-Bicocca, Vedano al Lambro, Italy. [9] Carbohydrate Signalling Group, Microbiology, School of Natural Sciences, University of Galway, Galway, Ireland. [10] These authors contributed equally: Renza Spelat, Federico Ferro, Paolo Contessotto. ✉email: abhay.pandit@universityofgalway.ie

Myocardial infarction (MI) is a common cause of cardiac injury in humans, resulting in irreversible loss of cardiomyocytes and the formation of fibrotic scar tissue that destabilizes the pump function, leading to congestive heart failure[1]. Consequently, patients have a lower quality of life and often die prematurely[2].

Many strategies have been explored to enhance cardiomyocyte proliferation, including biomolecular delivery, biomaterial augmentation and stem cell-based approaches[3,4], but these approaches do not compensate for the massive loss of cardiomyocytes.

In contrast to the adult mammalian heart, the zebrafish heart possesses an intrinsic capacity to regenerate upon myocardial injury through a robust proliferation of cardiomyocytes[5–8]. Ineffective cardiac regeneration in mammals is due to the low proliferative capacity of adult cardiomyocytes[8], which cannot dedifferentiate and re-enter the cell cycle as happens in zebrafish after a myocardial injury[8]. Interestingly, neonatal mice are also capable of regenerating myocardial damage induced by apical resection[9] or by left anterior descending artery ligation[10]. It has been demonstrated that the early postnatal changes in oxygen pressure, workload and availability of substrates for energy metabolism induce a metabolic shift from glycolysis to oxidative phosphorylation (OXPHOS)[11–13], triggering the exit of cardiomyocytes from the cell cycle shortly after birth[14,15]. An upregulation of glycolytic enzymes at the transcriptional level during cardiac regeneration has been reported in zebrafish[16]. Moreover, a decrease in cardiomyocyte proliferation after injury was detected when glycolysis was impaired. Interestingly, the manipulation of pyruvate metabolism using Pyruvate dehydrogenase kinase 3 (Pdk3) and a catalytic subunit of the pyruvate dehydrogenase complex proved the pivotal role of glycolysis on cardiomyocyte proliferation after injury[16]. Upregulation of genes that encode glycoenzymes and are regulators of pyruvate metabolism, such as Pdk2b, Pdk3b, and Pdk4 has been described in cell types surrounding damaged tissue[16,17]. Indeed, the cardiomyocytes in the border zone seemed to undergo a metabolic shift from oxidative phosphorylation to glycolysis and lactate fermentation. Such a metabolic reprogramming to an embryonic-like metabolic state was regulated through Neuregulin1/Receptor protein-tyrosine kinase ErbB2 (Nrg1/ErbB2) pathway[17]. Furthermore, the tumor suppressor gene Tumor protein 53 (Tp53), which has a role in cell cycle regulation, has been reported to be transiently suppressed by its negative regulator Mouse double minute 2 (Mdm2) during zebrafish heart regeneration[18].

Previous studies in neonatal mice revealed that new cardiomyocytes can derive from dedifferentiation and proliferation of pre-existing mature cardiomyocytes[10,19–22], similar to the zebrafish heart[8,23]. This would suggest that at least part of the mechanism underlying cardiomyocytes regenerative capacity could be analogous. Therefore, the cardiac regenerative capacity in zebrafish can be employed as a precious tool to further disclose the chain of processes that finally lead to heart regeneration.

Cell metabolism changes can induce modifications in glycosylation on the cell surface[24,25], which in turn play a fundamental role in most biological processes[26]. Indeed, our group has already described the distinctive pattern of membrane N- and O-linked glycosylation in neonatal mammalian heart[27]. Glycosylation is a dynamic process that influences both normal and pathologic processes, including cell motility and adhesion, cell–cell interactions, and immune system response[28,29]. However, our understanding of how N- and O-linked glycan structures are modified in the zebrafish heart regeneration model remains poor.

Thus, we examined transcriptomic, proteomic, metabolic and glycomic changes that occur in the ventricular tissue in a zebrafish cardiac regeneration model. Here, the combination of metabolic and glycomic profiles reflects the final response to autocrine and paracrine signaling and can actually advance our understanding of the mechanisms behind myocardial regeneration.

In this study, we performed the transcriptomic profile analysis identifying glycosylation and carbohydrate metabolism as key processes to regenerate the zebrafish heart. Consistently, proteomic analysis evidenced a metabolic shift from OXPHOS to glycolysis at seven days post-cryoinjury (dpci), which is an essential step to achieve cell cycle re-entry and heart regeneration[1]. In addition, glycosylation profile was compared between healthy and injured hearts at early timepoints post-cryoinjury focusing on specific cell surface N- and O-linked glycan structures by liquid chromatography-tandem mass spectrometry (LC-ESI-MS/MS), and revealed the complex cardiac remodeling. These findings are crucial for future efforts to design improved and effective methods for regeneration strategies after MI.

## Results

**Carbohydrate metabolism and glycosylation are key components of the transcriptional program in the early phase driving heart regeneration.** To induce cardiac damage and subsequent regeneration in zebrafish, we adopted the cryoinjury model which is based on rapid freezing-thawing of ventricular tissue. This model, in contrast to the resection-based one, induces a massive cell death followed by fibrotic tissue formation, resembling what happens in mammalian hearts after MI[30]. Specifically, we used a transgenic strain, where a green fluorescent protein gene was inserted under the promoter of cardiomyocyte-specific gene cardiac myosin light chain 2 (cmlc2)[31]. Thus, we were able to focus on cardiac regeneration process by following the proliferation of cardiomyocytes in the ischemic site (Supplementary Fig. 1). Three timepoints at specific days after cryoinjury (dpci) were taken into consideration: 2, 7 and 14 dpci.

First, we conducted bulk RNA-sequencing (RNA-seq) on ventricular tissue from each time point to investigate the transcriptional regeneration program. Sham vs injured samples were compared to define differentially expressed genes (DEG), and a 1% false discovery rate (FDR), an adjusted $p$ value ≤ 0.01, and fold change (FC) > 1.5 log$_2$ were applied as threshold settings. Using these criteria, we identified 3889 DEG at 2 dpci, 1455 at 7 dpci and 2277 at 14 dpci (Supplementary Figs. 2 and 3 and Supplementary Data 1–3).

Next, we integrated expression data with functional analyses. In particular, Database for Annotation, Visualization, and Integrated Discovery (DAVID)[32] was used to perform a gene-annotation enrichment analysis of the set of DEG (adjusted $p$ value < 0.01), while the R package GOplot 1.0.2[33] was used to visualize the analysis. The results evidenced that "carbohydrate metabolic processes", "Arp2/3 complex-mediated actin nucleation", influencing cell migration[34], "neutrophil chemotaxis" and "N-linked glycosylation" were among the functional categories that were highly enriched at 2 dpci, and that most of the genes belonging to these functional categories were upregulated (Fig. 1a and Supplementary Table 1). At 7 dpci, "carbohydrate metabolic processes" and "inflammatory response" were still the most activated functions (Fig. 1b and Supplementary Table 2), while at 14 dpci, "protein folding" and processes related to protein synthesis such as "rRNA processing" and "rRNA biogenesis" were the most activated. Interestingly, the category "carbohydrate metabolic processes" was still one of the most enriched and activated, and "N-linked glycosylation" was seen as a significant function at all the three timepoints (Fig. 1c and Supplementary Table 3). These results highlight the relevance of carbohydrate metabolism and glycosylation in the heart regeneration process.

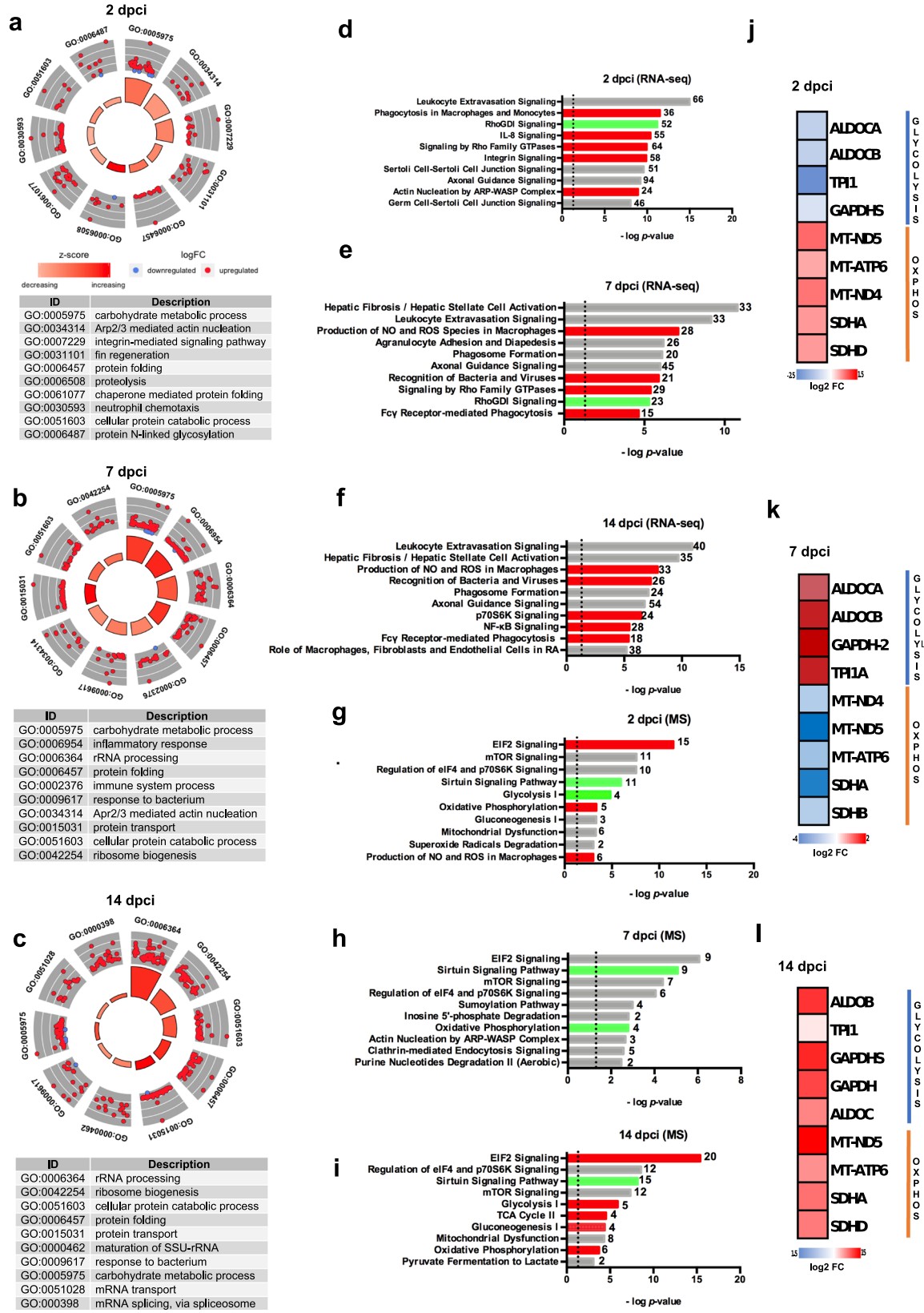

To further define the regeneration transcriptional program, we performed Ingenuity Pathway Analysis (IPA®), identifying the most relevant pathways associated with each time point. At 2 dpci, we observed an induction of pathways which are linked to inflammation response, such as "phagocytosis in macrophages and monocytes", and cell migration, such as "signaling by Rho family GTPases" (Fig. 1d). The relevance of inflammation response at this stage of regeneration was also confirmed by the high expression of CD163 (Supplementary Fig. 4b, c), a macrophage marker used to identify regulatory macrophages[35].

At 7 dpci, the prominence of fibrosis and tissue remodeling in the damaged zone were highlighted by the presence of the

**Fig. 1 The response to injury is characterized by molecular and metabolic changes. a** GOCircle plot summarizing gene ontology enrichment analysis at 2 dpci, **b** 7 dpci and **c** 14 dpci; "carbohydrate metabolic process" and "N-linked glycosylation" are identified as significantly activated processes, among others. The significance of each term (-$\log_{10}$ adjusted $p$ value) is specified by the height of the bar plot in the inner ring, while the color corresponds to the z-score. The expression level ($\log_2$FC) for the genes in each term is displayed in the outer ring scatterplots; red dots indicated upregulated genes and blue dots downregulated genes. **d** The most statistically significant canonical pathways for the transcriptomic analysis at 2 dpci, **e** 7 dpci and **f** 14 dpci identified using the IPA® software are listed according to their $p$ value (-log); the threshold -log($p$ value) = 1.3 corresponds to a $p$ value of 0.05. Activation is indicated in red, inhibition in green and unpredictability in gray. **g** The most statistically significant canonical pathways for the proteomic analysis at 2 dpci, **h** 7 dpci and **i** 14 dpci. **j** Heatmap showing glycolytic enzymes and mitochondrial oxidative phosphorylation proteins at 2 dpci, **k** 7 dpci and **l** 14 dpci. Two biological replicates deriving from the pooling of 4 different animals ($n = 8$ animals per group) were analyzed for all experiments. For proteomic analysis $n = 4$ animals per group were used.

"Hepatic fibrosis/Hepatic stellate cell activation" as the most significant pathway (Fig. 1e), and by the higher accumulation of both vimentin and smooth muscle actin (Supplementary Figs. 4e, f and 5), reported to be markers for post-MI fibroblasts in zebrafish[36]. The fibrosis observed was transient, as previously reported[37], and its resolution and replacement with cardiac tissue was evident at 14 dpci, with activation of pathways linked to cardiomyocytes cell cycle re-entry and proliferation, such as "NF-$k$B signaling"[38] (Fig. 1f), and GATA4 nuclear localization (Supplementary Fig. 4h, i), which are essential for cardiomyocyte dedifferentiation[39]. Comparison analysis of the pathways identified in all the timepoints evidenced a more sustained activation at 7 dpci (Supplementary Fig. 4j).

We also performed an upstream regulator analysis (URA), that permits the identification of the molecules upstream of the genes in the dataset and their associated networks[40]. We found that many cytokines associated with the inflammatory response, including colony-stimulating factor 2 (CSF2), tumor necrosis factor (TNF), interferon-gamma (IFNG), interleukin 6 (IL-6), and interleukin 4 (IL-4)[41], and transcriptional regulators related to blood vessel formation, including vascular endothelial growth factor A (VEGFA), epidermal growth factor (EGF), fibroblast growth factor (FGF), and insulin-like growth factor (IGF)[42] were activated throughout the entire period studied, with an increasing significance of anti-inflammatory (IL-13, IL-4)[43] and pro-angiogenic (VEGFA, EGF) functions starting from 7 dpci (Supplementary Fig. 4a, d, g).

Previously reported RNA-seq datasets partially agree with our findings, and the dissimilarities may be due to the wide variability among the aim, experimental design, strain and timepoints studied (Supplementary Table 4 and Supplementary References 1–8).

**Metabolic switch from mitochondrial oxidative phosphorylation to glycolysis characterizes heart regeneration at 7 dpci.** Functional analyses on RNA-seq data identified carbohydrate metabolism and glycosylation as critical functional categories during heart regeneration, and it also enabled the assignment of specific regeneration stages by considering the major biological functions identified at each time point; 2 dpci named as the "inflammation phase", 7 dpci as the "fibrosis and re-vascularization phase", and 14 dpci as the "cardiomyocytes proliferation phase".

Moreover, to characterize the proteome at each time point, we used a combination of high-performance nano-liquid chromatograph (LC) and tandem mass spectrometry (MS/MS). We identified and quantified 262 differentially expressed proteins (1% FDR and adjusted $p$ value ≤ 0.01) (Supplementary Data 4). IPA® pathway analysis was performed on these differentially expressed proteins and evidenced a metabolic reprogramming through heart regeneration phases, primarily involving mitochondrial OXPHOS and glycolysis. Briefly, glycolysis is composed of 10 enzymatic steps, through which glucose is metabolized to pyruvate. Subsequently, glycolytic pyruvate enters into the

mitochondrial tricarboxylic acid cycle (TCA) to generate reduced nicotinamide adenine dinucleotide (NADH) and flavin adenine dinucleotide (FADH2), which are utilized by mitochondrial OXPHOS complexes I–V to generate adenosine triphosphate (ATP)[44]. Interestingly, 2 dpci, the "inflammation phase", was characterized by an activation of mitochondrial OXPHOS and an inhibition of glycolysis (Fig. 1g and Supplementary Fig. 4l). This result was not unexpected, since adult cardiomyocytes rely on a fatty acid metabolism and aerobic mitochondrial OXPHOS rather than on glycolysis for their energy production[45].

Importantly, at 7 dpci, the "fibrosis and re-vascularization phase", we identified a metabolic switch from mitochondrial OXPHOS to glycolysis, as demonstrated by IPA® pathway analysis (Fig. 1h and Supplementary Fig. 4l).

Finally, the "cardiomyocytes proliferation phase", coupled to 14 dpci, was marked by a simultaneous activation of mitochondrial OXPHOS and glycolysis pathways (Fig. 1i and Supplementary Fig. 4l).

Consistent with these findings, many glycolytic enzymes were downregulated compared to sham at 2 dpci, such as fructose-bisphosphate aldolase C-A (ALDOCA; $\log_2$FC = −1), fructose-bisphosphate aldolase C-B (ALDOCB; $\log_2$FC = −1), triose-phosphate isomerase (TPI1; $\log_2$FC = −2.4), and glyceraldehyde-3-phosphate dehydrogenase-S[45] (GAPDHS; $\log_2$FC = −1), while proteins involved in mitochondrial OXPHOS, such as mitochondrial NADH-ubiquinone oxidoreductase chain 4 (MT-ND4; $\log_2$FC = 1.1), mitochondrial NADH-ubiquinone oxidoreductase chain 5 (MT-ND5; $\log_2$FC = 1.2), mitochondrial ATP Synthase Membrane Subunit 6 (MT-ATP6; $\log_2$FC = 1), succinate dehydrogenase complex flavoprotein subunit A (SDHA; $\log_2$FC = 1), succinate dehydrogenase complex flavoprotein subunit D (SDHD; $\log_2$ FC = 1), were upregulated (Fig. 1j). On the contrary, at 7 dpci we observed an upregulation of the glycolytic enzymes ALDOCA ($\log_2$FC = 1.4), ALDOCB ($\log_2$FC = 2), TPI1A ($\log_2$FC = 2), GAPDH-2 ($\log_2$FC = 2.3) associated with downregulation of the mitochondrial OXPHOS proteins MT-ND4 ($\log_2$FC = −1), MT-ND5 ($\log_2$FC = −4), MT-ATP6 ($\log_2$FC = −1), SDHA ($\log_2$FC = −2.6), SDHB ($\log_2$FC = −1) (Fig. 1k). At 14 dpci, the upregulation of both glycolytic enzymes ALDOCB ($\log_2$FC = 3.9), ALDOC ($\log_2$FC = 2.4), TPI1A ($\log_2$FC = 0.5), GAPDH ($\log_2$FC = 3.6), GAPDHS ($\log_2$FC = 4.2), and OXPHOS proteins MT-ND5 ($\log_2$FC = 4.9), MT-ATP6 ($\log_2$FC = 2.1), SDHA ($\log_2$FC = 2.7), SDHD ($\log_2$FC = 2.5) further confirmed the simultaneous activation of mitochondrial OXPHOS and glycolysis pathways (Fig. 1l).

Based on the previous cell-type computational analysis on bulk RNA-seq data evidencing a high proportion of cardiomyocytes (Supplementary Fig. 4k), we decided to address energy metabolism changes in cardiomyocytes using targeted spatial sequencing. In particular, we analyzed the levels of *gapdh* and *pkma* (pyruvate kinase M1/2a), belonging to the glycolysis pathway, *mt-co1* (Mitochondrially Encoded Cytochrome C Oxidase I) and *sdha*,

involved in the OXPHOS pathway, for each time point taken into consideration (Fig. 2a–d and Supplementary Fig. 6).

Since the analysis was performed on the transgenic zebrafish line cmlc2, expressing gfp under the control of cmlc2 promoter, gfp was chosen as a cardiomyocyte marker. The expression of gapdh, pkma, mt-co1 and sdha was compared between cardiomyocytes of the border zone (BZ) and those of the remote zone (RZ).

The results evidenced an increase of gapdh (2 FC) and pkma (2 FC) in BZ cardiomyocytes compared to RZ ones at 7 dpci. At 2 and 14 dpci the expression levels were comparable in these two zones (1.1 FC and 1.2 FC, respectively). On the contrary, we observed an increase in mt-co1 expression at 2 dpci (1.7 FC), and a decrease at 7 dpci (−2.3 FC) in BZ cardiomyocytes as compared to RZ cardiomyocytes, while at 14 dpci the expression levels were comparable (1.1 FC). Similarly, sdha increased at 2 dpci (2 FC), decreased at 7 dpci (−2 FC) and was similar at 14 dpci (1.2 FC). Interestingly, we evidenced a simultaneous induction of glycolysis and OXPHOS genes at 14 dpci, as compared to 7 dpci, both in BZ (gapdh 3 FC; pkma 2 FC; mt-co1 6 FC; sdha 4 FC) and RZ (gapdh 4 FC; pkma 3 FC; mt-co1 5 FC; sdha 5.4 FC) cardiomyocytes (Fig. 2e).

Taken together, these results highlight a metabolic reprogramming that is characterized by a switch from mitochondrial OXPHOS to glycolysis at 7 dpci in BZ cardiomyocytes, and a concomitant activation of both metabolic pathways at 14 dpci, during the "cardiomyocytes proliferation phase", involving both BZ and RZ cardiomyocytes. This metabolic program differs from the physiological one, which relies on mitochondrial OXPHOS for energy production, thereby demonstrating its implication in zebrafish heart regeneration.

**Alterations in cell membrane sialylation characterize the inflammatory phase.** To further investigate glycosylation during the heart regeneration process, which was indicated as altered by the transcriptomic analysis, we carried out three types of glycosylation analysis on cell membrane glycans[46]. These glycomic experiments include (i) lectin microarray, to investigate variation in global protein glycosylation, (ii) lectin histochemistry, to localize specific glycan residues within the tissue, and (iii) LC-ESI-MS/MS, to analyze glycan structures[46].

The lectin microarray was composed of a panel of 49 immobilized lectins with established binding specificities (Supplementary Table 5). The resulting binding intensity data indicated the presence of high mannose structures, suggested by binding to Lens culinaris (Lch-B), Calystegia sepium (Calsepa), Narcissus pseudonarcissus (NPA), Galanthus nivalis (GNA) and Canavalia ensiformis (Con A) lectins as well as to glycans containing sialic acids, as demonstrated by binding to Trichosanthes japonica (TJA-I), Maakia amurensis (MAA), Sambucus nigra (SNA-I) and Triticum vulgaris (WGA) lectins (Fig. 3a and Supplementary Table 5). Changes in the abundance of glycan motifs across sample conditions, i.e., the specific regeneration phase, in respect to the uninjured heart (sham) were also evaluated.

In particular, at 2 dpci, the "inflammatory phase", we observed a significant decrease in binding of: (i) MAA lectin ($\log_2$ FC = −0.4), indicating a reduction in α-(2,3)-linked sialylation; (ii) WGA ($\log_2$ FC = −0.5), which recognizes GlcNAc and/or sialic acid residues; (iii) Lotus tetragonolobus (LTA, $\log_2$ FC = −0.4) and Ulex europaeus (UEA-I, $\log_2$ FC = −0.3) lectins, with binding preferences for fucose (Fuc) in different linkages (LTA for α-(1,3)-, α-(1,6)- and α-(1,2)-linked Fuc, and UEA-I for α-(1,2)-linked Fuc only); and (iv) Robinia pseudoacacia (RPbAI,

$\log_2$ FC = −0.5), a galactose (Gal) and GalNAc binding lectin (Fig. 3a, b, Supplementary Table 5 and Supplementary Data 5).

At 7 dpci, the "fibrosis and neo-vascularization phase", in addition to MAA ($\log_2$ FC = −0.5), LTA ($\log_2$ FC = −0.5) and UEA-I ($\log_2$ FC = −0.5), there was also a decrease in the binding of Con A lectin ($\log_2$ FC = −0.3), which binds to α-linked mannose (Man), Glc and GlcNAc, and Lch-A ($\log_2$ FC = −0.5), and recognizes Man and Glc, both indicating the presence of high mannose-type glycan structures; Artocarpus integrifolia (AIA, $\log_2$ FC = −0.6), showing affinity for Gal, Gal-β-(1,3)-GalNAc and tolerance to sialylation; and Glycine max (SBA, $\log_2$ FC = −0.2), which binds GalNAc residues (Fig. 3a, c, Supplementary Table 5 and Supplementary Data 6).

At 14 dpci, the "cardiomyocytes proliferation phase", the profile was very similar to that of the sham, showing a decrease only in RPbAI binding ($\log_2$ FC = −0.4) (Fig. 3a, d and Supplementary Data 7).

Taken together, these results suggest a complex membrane glycan remodeling taking place during the different phases of regeneration, with a reduction of α-(2,3)-sialylation, fucosylation and galactosylation in both the "inflammation phase" and the "fibrosis and neo-vascularization phase". Moreover, the "fibrosis and neo-vascularization phase" was also marked by a decrease in high mannose structures. Conversely, a reconstitution of the healthy (i.e., uninjured) heart glycosylation was achieved in the "cardiomyocytes proliferation phase", which shows a reduction in β-linked galactosylated residues when compared to the sham.

In addition, hierarchical clustering analysis identified two main clusters, one cluster including sham, 7 and 14 dpci samples, and the other cluster containing sham biological replicate 2 (Sham_2), 7 dpci biological replicate 3 (7 dpci_3), and all 2 dpci samples. With the exception of the Sham_2 and 7 dpci_3 samples, which represent individual and/or biological variability, sham and 14 dpci samples were included in the same cluster, indicating a similarity of global protein glycosylation. This supports the previous transcriptomic data which indicated a characteristic biological behavior associated with each group. All 2 dpci samples clustered together, suggesting that they have greater differences in overall glycosylation than those of 7 dpci, 14 dpci and sham samples, and that the greatest glycosylation disruption occurred early post-cryoinjury with conditions returning to more "normal" or healthy glycosylation after that time point (Fig. 3a).

Given the relevance of sialylation in many physiological and pathological processes, such as inflammation, cardiovascular diseases and embryogenesis[47], we investigated sialic acid distribution within the myocardial tissue using lectin histochemistry.

In particular, we selected three lectins to recognize different sialic acid linkages: MAA, which is specific for terminal α-(2,3)-linked sialic acid; SNA-I, which recognizes terminal α-(2,6)-linked sialic acid; and WGA, which binds to sialic acid without linkage preference and GlcNAc residues. For all these lectins binding was mainly observed in pre-existing and new-forming vessels that can be easily identified based on their shape, indicating the presence of both terminal α-(2,3)- and α-(2,6)-linked sialic acids on these tissue structures (Fig. 4a–c), even though conclusions on the specific cell types cannot be made. Binding intensity quantification indicated a significant decrease in MAA binding at 2 dpci compared to sham, in line with lectin microarray data, while at 7 dpci we observed an increase of MAA binding intensity, which was similar to that in sham tissues by 14 dcpi (Fig. 4d). The same trend was seen for SNA-I and WGA lectin binding (Fig. 4e, f), suggesting an overall decrease in α-(2,3)- and α-(2,6)-sialylation at 2 dpci, followed by a gradual re-establishment of the normal sialic acid quantity in the following phases.

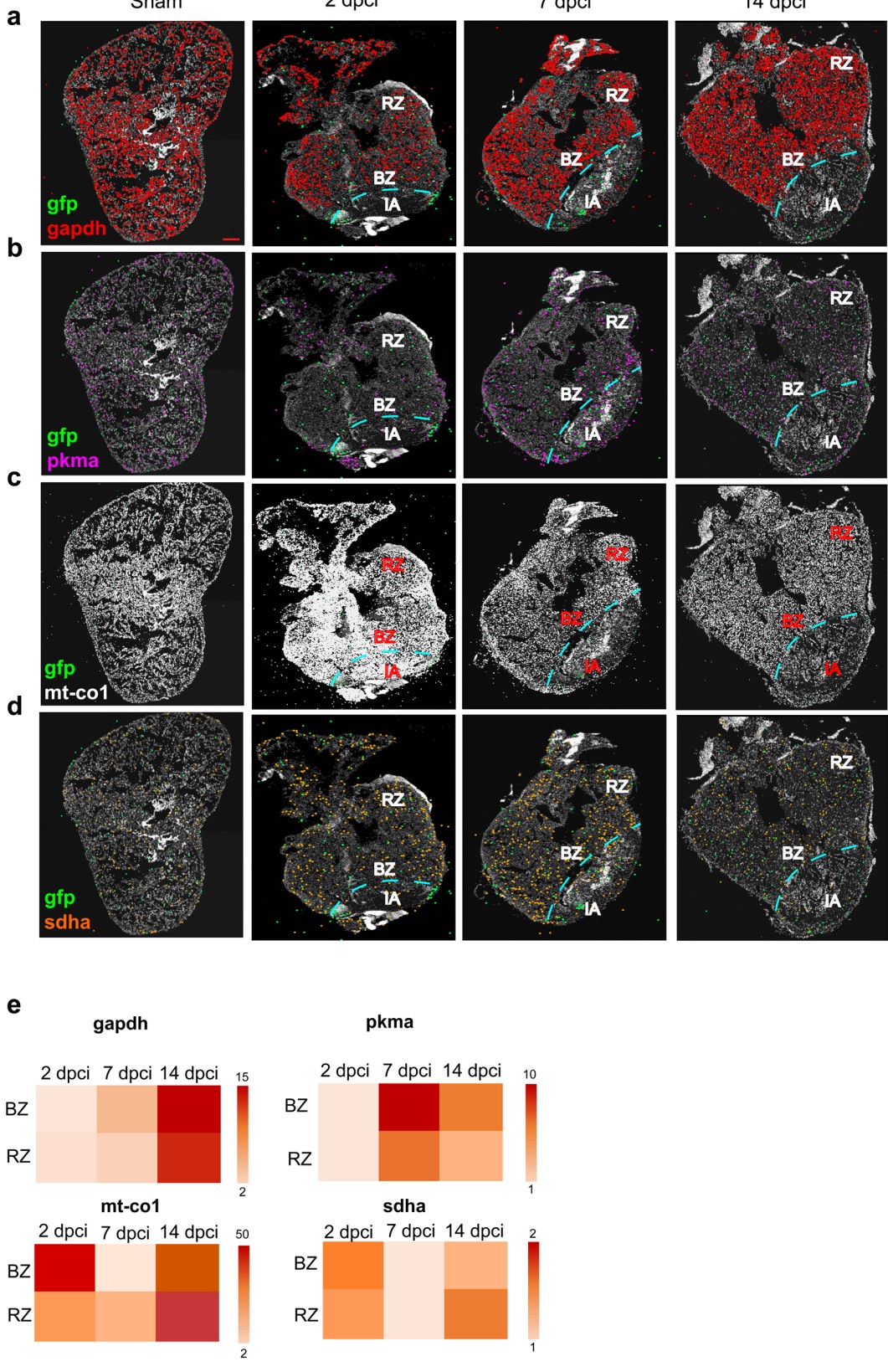

**Fig. 2 Cardiomyocyte targeted in situ sequencing.** A targeted multiplexed mRNA detection assay employing padlock probes, rolling circle amplification (RCA), and barcode sequencing was applied to detect the spatial distribution of **a** *gapdh* **b** *pkma* **c** *mt-co1* and **d** *sdha* in sham and at 2, 7 and 14 dpci. All the experiments were performed on *cmlc2* zebrafish transgenic strain, where a green fluorescent protein (*gfp*) gene was inserted under the promoter of cardiomyocyte-specific gene cardiac myosin light chain 2. For this reason, *gfp* was chosen as a cardiomyocyte marker. IA injured area, BZ border zone, RZ remote zone. Scale bar = 100 µm. **e** Heatmaps showing the expression comparison of *gapdh*, *pkma*, *mt-co1* and *sdha* between BZ and RZ at each time point. Heat legend represents the relative quantification of the spot number and it is represented as fold change. Two biological replicates were analyzed.

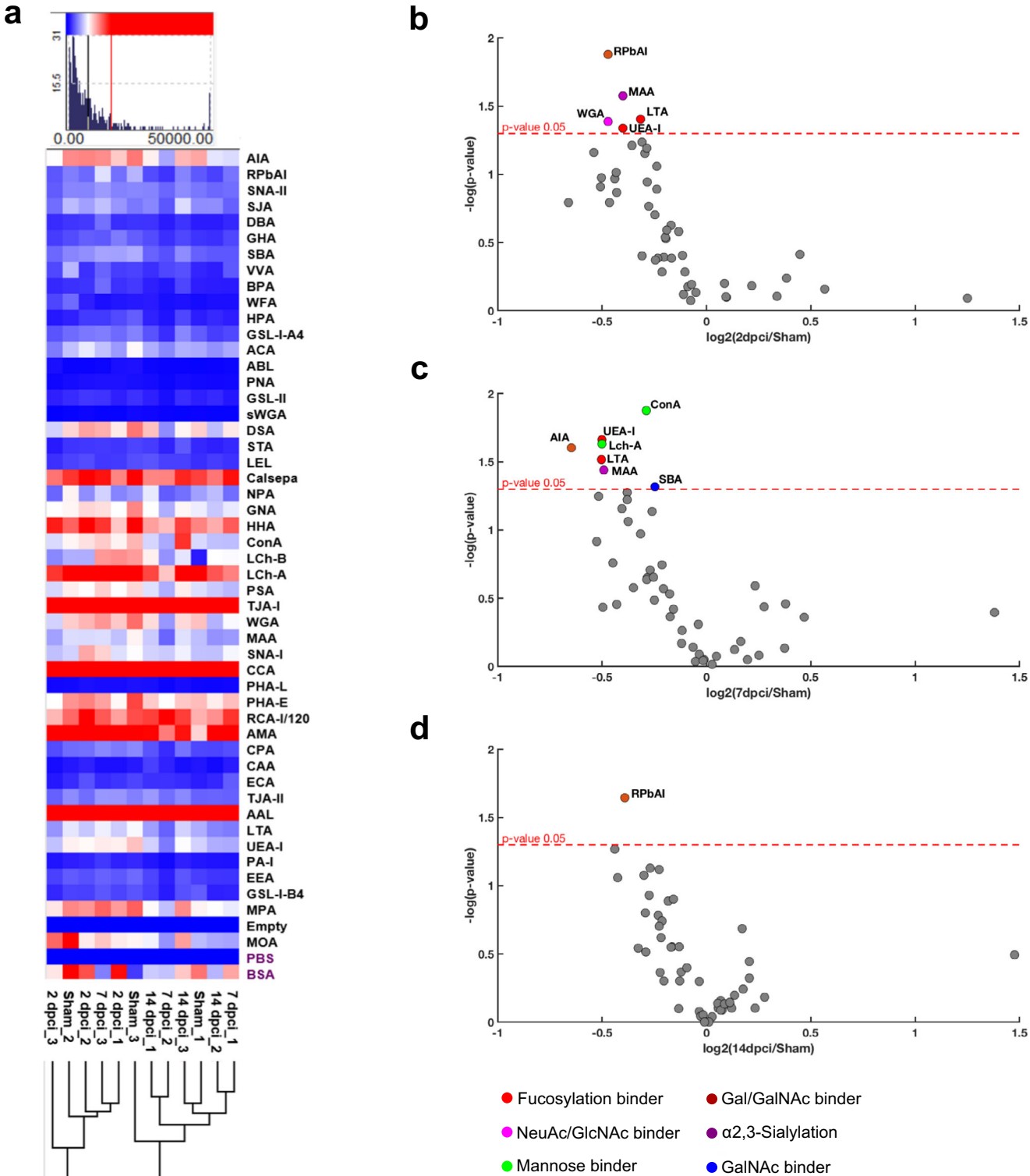

**Fig. 3 Lectin microarray analysis evidences cell membrane sialylation alterations in zebrafish heart during regeneration. a** Hierarchical clustering of lectin microarray data. Asialofetuin (ASF) was used as control glycoprotein and all lectin binding intensities were normalized. **b** Lectin microarray dot plot of 2 dpci, **c** 7 dpci and **d** 14 dpci in respect to the sham showing a reduction of: sialylation (MAA, WGA), fucosylation (LTA, UEA-I), GlcNAc (WGA) and Gal/GalNAc (RPbAI) at 2 dpci; sialylation (MAA), fucosylation (LTA, UEA-I), mannose (Con A, Lch-A), Gal/GalNAc (AIA) and GalNAc (SBA) at 7 dpci; and Gal/GalNAc (RPbAI) at 14 dpci. One-way ANOVA analysis with Tukey's post hoc correction was applied. Spot colors correspond to lectin sugar specificity; the red dotted line represents a significant cutoff of $p$ value $\leq 0.05$. Three biological replicates were analyzed, each one deriving from the pooling of 4 different animals ($n = 12$).

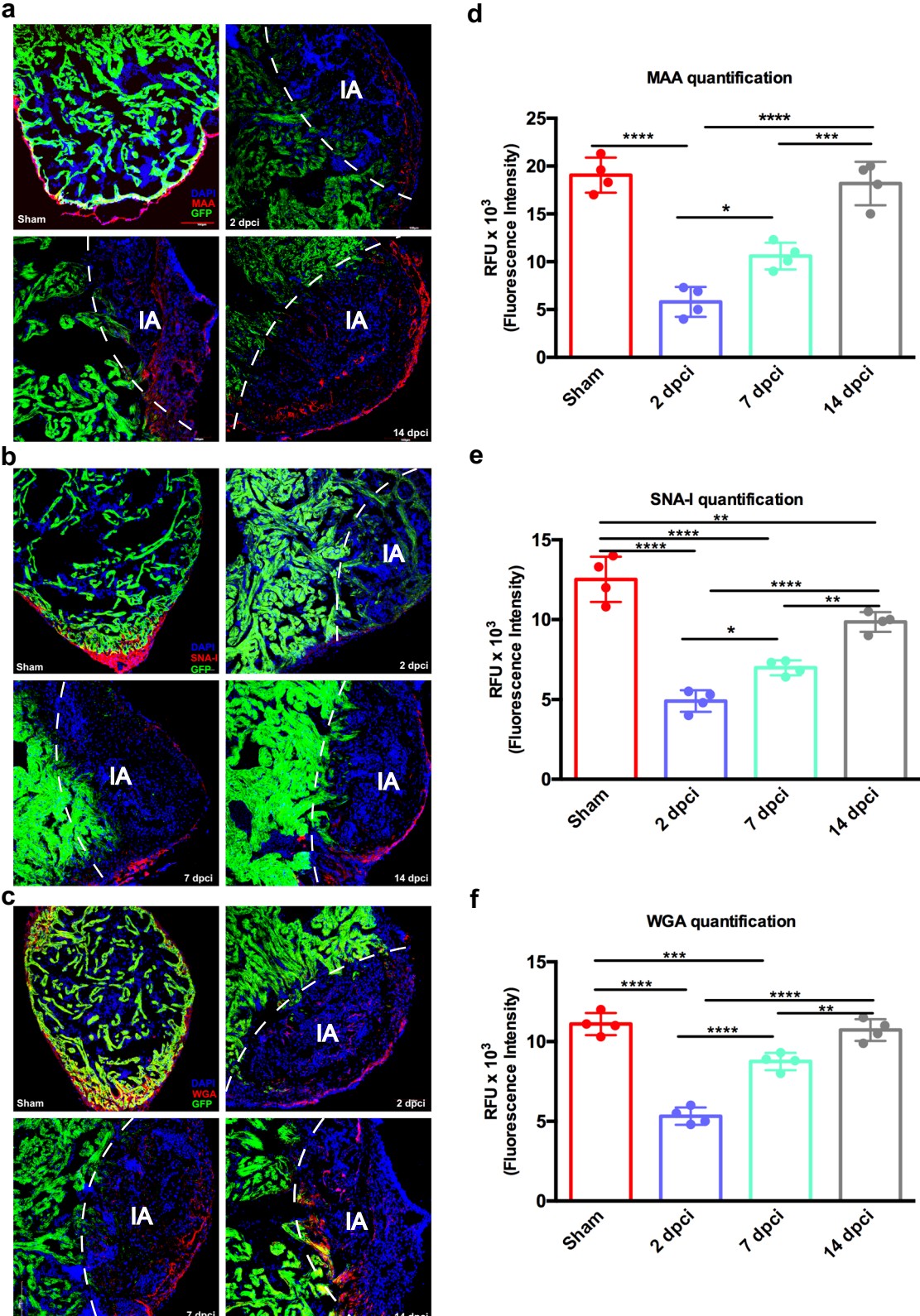

**Fig. 4 Sialic acid distribution in regenerating zebrafish heart identified by lectin histochemistry. a** Lectin histochemistry showing the distribution of MAA, recognizing α-(2,3)-linked sialic acid, **b** SNA-I, binding to α-(2,6)-linked sialic acid and **c** WGA, binding to sialic acid and GlcNAc, in ventricle tissue. **d** MAA, **e** SNA-I and **f** WGA relative quantification. The lectin binding quantifications highlighted a similar decrease at 2 dpci and a subsequent increase in the following timepoints. Cardiomyocytes were identified by GFP positivity. IA injured area. Data are shown as mean with SD ($n = 4$ animals per group). Scale bar = 100 µm. $*p < 0.05$, $**p < 0.01$, $***p < 0.001$, $****p < 0.0001$.

**Cell membrane N- and O-glycans remodeling during heart regeneration**. To further investigate glycan structural changes during heart regeneration, N-linked glycans and O-linked glycans released from membrane proteins were analyzed by LC-MS/MS. In total, we detected 76 unique N-glycan structures (NG1-76) (Supplementary Data 8), with an overall prevalence of high mannose structures compared to hybrid and complex type, regardless of the regeneration phase (Fig. 5a, e). In particular, the terminal disaccharide Neu5Ac-α-(2,8)-Neu5Ac-α-(2,3)-R (diSia) structure and galactosylated Lewis x structures (Gal-β-(1,4)-Gal-β-(1,4)-[Fuc-α-(1,3)-]GlcNAc-β-R) with or without sialylation, which were previously reported in zebrafish embryoes[48,49], were also detected (Supplementary Data 8).

The relative percentage of high mannose structures decreased from 53% in sham to 47% at 2 dpci, and this decrease was mainly due to a 5% reduction (from 11% in sham to 6% at 2 dpci) of the Hex6HexNAc2 structure (Man6, NG7 in Supplementary Data 8), that progressively increased at 7 dpci (7%) and 14 dpci (11%) (Fig. 5c and Supplementary Data 8). Consistently with previous results from lectin microarrays, a significant reduction in the binding of Con A and Lch-A, which both bind to mannose, was seen at 7 dpci, but not at 2 dpci, compared to sham (Fig. 3a, b and Supplementary Data 5 and 6).

Moreover, the relative percentage of complex type N-glycans increased from sham (38%) to 2 dpci (43%), then slightly decreasing at 7 and 14 dpci (38% and 39%, respectively). The relative percentage of hybrid type N-glycans was quite stable (9% in Sham, 10% in 2 dpci, 9% in 7 dpci, and 11% in 14 dpci), also focusing only on sialylated hybrid structures (sham 7%, 2 dpci 7%, 7 dpci 6%, 14 dpci 6%) (Fig. 5a, b and Supplementary Data 8). Specifically, the N-acetylneuraminic acid (Neu5Ac) quantity was comparable to sham at 7 and 14 dpci (28%), but was increased at 2 dpci (33%). Similarly, N-glycolylneuraminic acid (Neu5Gc) slightly increased only at 2 dpci (3%) compared to the other timepoints (2%). There was an overall increase of 6% in sialylated complex and hybrid type N-glycans at 2 dpci (sham 30%, 2 dpci 36%, 7 dpci 30%, 14 dpci 30%), with the α-(2,3)-linkage type largely contributing to this increase (5%). Interestingly, all α-(2,6)-sialylated structures were identified only in low amounts at 2, 7 and 14 dpci, and not in the Sham (Fig. 5d and Supplementary Data 8). In contrast, lectin microarray and histochemistry results indicated an overall decrease in α-(2,3)- and α-(2,6)-sialylation at 2 dpci, suggesting that sialylated N-glycan structures detected by LC-MS/MS did not contribute substantially to the changes observed, and that sialylated O-linked glycans could account for this difference, as it will be shown later.

The O-linked glycan structural analysis identified 29 O-glycans (OG1-29) comprising core 1 and 2 type structures. Core 1 O-glycans were more abundant (82–98%) than core 2 (2–18%) in all conditions. Interestingly, the relative abundance of core 1 structures decreased from sham (98%) to 2 dpci (82%), 7 dpci (93%) and 14 dpci (94%) (Fig. 6a, c and Supplementary Data 9), and these results were consistent with lectin microarray data where a tendency in a reduced binding of AIA lectin was seen at 2 and 14 dpci compared to sham (Fig. 3 and Supplementary Data 5–7). The abundance of core 2 structures increased from sham (2%) to 2 dpci (18%), 7 dpci (7%) and 14 dpci (6%) (Fig. 6a, c and Supplementary Data 9).

Interestingly, all identified structures were sialylated in the sham (100%), while sialylation was decreased in 2 dpci to 88%, and subsequently increased in 7 dpci to 93% and 14 dpci to 98%. This reduction in sialylation was due to core 1 structures, while core 2 sialylated structures increased from sham (2%) to 2 dpci (10%) (Fig. 6a–c and Supplementary Data 9). Importantly, the overall decrease in sialylated O-glycans at 2 dpci was mainly due to a reduction in disialylated structures, containing both α-(2,3)-

and α-(2,6)-linked sialic acids, thus supporting the lectin microarray and histochemistry results of reduced binding of the lectins MAA and SNA-I. The reduction in WGA binding was also consistent with the LC-MS/MS results (Figs. 3a, 4, and 6b, d and Supplementary Data 9).

Non-sialylated core 1 structures were absent in the sham, and were present at 4% in 2 dpci, 5% in 7 dpci and 2% in 14 dpci. Similarly, non-sialylated core 2 structures were not detected in the sham, and were present at 8% in 2 dpci and 2% in 7 dpci, but were absent in 14 dpci. Moreover, two sulfated core 1 O-glycans were detected only in injured samples, (6S)Gal-β-(1,3)-GalNAcol structure was present only in 2 dpci (1%), and Neu5Ac-α-(2,3)-(6S)Gal-β-(1,3)-GalNAcol in 2 dpci (1%), 7 dpci (2%) and 14 dpci (2%) (Supplementary Data 9).

## Discussion

Zebrafish heart regeneration is very efficient and relies on the proliferation of pre-existing cardiomyocytes[17], but in mammals this capability is limited to the early neonatal stage[11]. Elucidating the mechanisms associated with this repair potential and regenerative capacity can contribute to the discovery of novel therapeutic targets for cardiac diseases. In this study, we focused on two poorly explored biological processes: carbohydrate metabolism and glycosylation, demonstrating their profound alterations in the crucial phases of heart regeneration. Our findings suggest that metabolic regulation and membrane glycosylation remodeling play a critical role in driving zebrafish heart regeneration.

We first investigated the transcriptional changes occurring during zebrafish heart regeneration using a deep sequencing approach. Enrichment analysis of DEG revealed time point-specific responses, identifying GO terms related with immune processes, such as "neutrophil chemotaxis" and "inflammatory response", as the most enriched at 2 dpci and 7 dpci, respectively.

Previous studies have demonstrated that the immune response plays a critical role in initiating tissue regeneration[50–53], and our results are in agreement with these findings, confirming the relevance of the immune processes in the early phases of regeneration. GO terms related to protein synthesis were among the most enriched at 14 dpci, indicating that the cells are prepared to create a new set of proteins that, when combined with extensive protein degradation, will lead to a regenerative-like cellular proteome[54].

Interestingly, the GO terms "carbohydrate metabolic process" and "protein N-linked glycosylation" were enriched in all the timepoints analyzed, suggesting the importance of metabolic regulation and glycan remodeling during the regeneration process. Indeed, proteomic analysis identified a metabolic switch from mitochondrial OXPHOS to glycolysis immediately after the "inflammatory phase" (2 dpci), and simultaneous activation of both metabolic pathways in the "cardiomyocytes proliferation" phase (14 dpci). Previous studies exploiting cardiac regeneration in neonatal mice revealed that new cardiomyocytes can derive from the dedifferentiation and proliferation of pre-existing mature cardiomyocytes, similarly to what happens during zebrafish heart regeneration[10,19–22]. Moreover, it has been demonstrated that the first week of mammalian postnatal heart development is characterized by a metabolic shift from glycolysis to OXPHOS, because of physiological adaptations to changes in oxygen pressure, workload and availability of substrates for energy metabolism[11,12,14]. This increase in oxygen supply induces the subsequent upregulation of the oxidative metabolism, as well as a rise in the production of mitochondrial reactive oxygen species (ROS), causing cardiomyocytes to exit the cell cycle shortly after birth[14,15]. In particular, proliferating cardiomyocytes

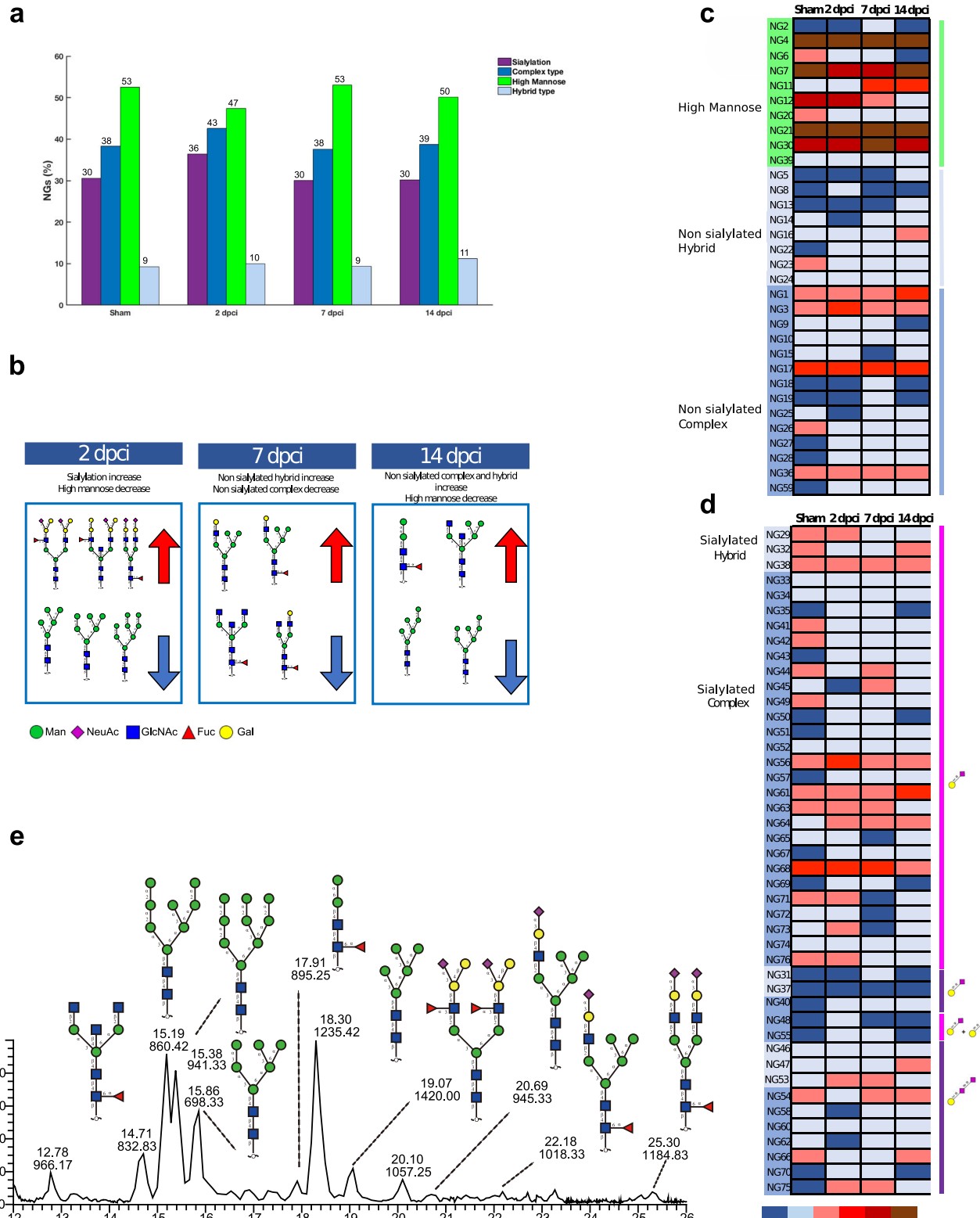

**Fig. 5 Membrane N-glycan profile of regenerating zebrafish heart established by LC-MS/MS revealed increased sialylation and a reduction in high mannose structures during the inflammatory phase. a** Comparison of N-glycan types (complex, high mannose and hybrid) and sialylation between sham and the different regeneration phases (2, 7 and 14 dpci). The inflammatory phase (2 dpci) showed a 6% increase in sialylated glycans and a 5% decrease in high mannose structures compared to sham. **b** Summary of the main changes in N-glycans features (sialylated hybrid and complex type increase/decrease, non-sialylated hydrid and complex type increase/decrease, high mannose-type increase/decrease) showing some of the corresponding structures. **c** Heatmap displaying time-specific relative quantification of neutral, and **d** sialylated N-glycans. Color intensity represents the relative abundance expressed as a percentage. **e** Extracted ion chromatography (EIC) showing the most abundant membrane N-glycans present in zebrafish heart. Results deriving from samples pooling of six animals are shown ($n = 6$ animals per group).

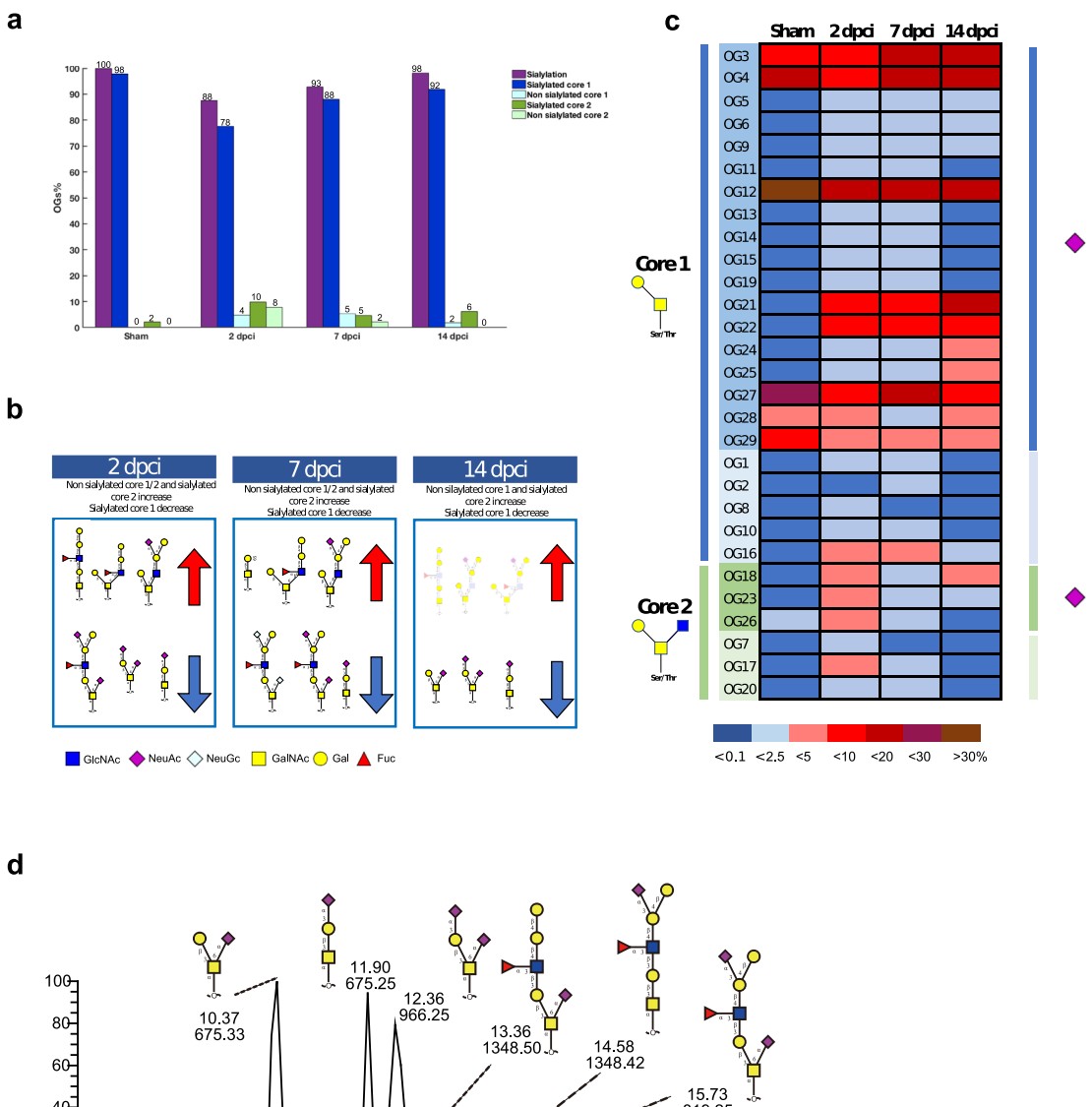

**Fig. 6 Membrane O-glycan profile of regenerating zebrafish heart established by LC-MS/MS revealed decreased sialylation during the inflammatory phase. a** Comparison of O-glycan structures (core 1 and core 2) and sialylation between sham and the different regeneration phases (2, 7 and 14 dpci). The inflammatory phase (2 dpci) showed a 12% decrease in sialylated O-glycans and a subsequent increase at 7 and 14 dpci. **b** Summary of the main changes in O-glycan structures (sialylated core 1/2 increase/decrease, non-sialylated core 1/2 increase/decrease) showing some of the corresponding structures. **c** Heatmap displaying time-specific relative quantification of core 1 and core 2 O-glycans. Sialylated structures are also indicated (purple square). Color intensity represents the relative abundance expressed as a percentage. **d** EIC showing the most abundant membrane O-glycans present in zebrafish heart. Results deriving from samples from the pooling of six animals are shown ($n = 6$ animals per group).

depend on glycolysis in postnatal day 1 (P1) hearts, while in postnatal day 7 (P7) cardiomyocytes rely on OXPHOS, as the adult[11]. Here, by focusing on the molecules involved in energy metabolism, we found that many enzymes assigned to glycolysis were downregulated at 2 dpci in the zebrafish heart, whereas OXPHOS enzymes were upregulated. Interestingly, the opposite was observed at 7 dpci, demonstrating a metabolic shift from OXPHOS to glycolysis.

To address metabolic changes specifically occurring in cardiomyocytes, we performed targeted in situ sequencing, comparing BZ (border zone) and RZ (remote zone) cardiomyocytes. A metabolic reprogramming from OXPHOS to glycolysis in BZ cardiomyocytes was observed at 7 dpci, indicating the activation of an embryonic phenotype due to dedifferentiation processes as previously demonstrated[17]. Interestingly, here we observed for the first time that, after the metabolic shift to glycolysis in the BZ, cardiomyocytes from both BZ and RZ exhibit a simultaneous induction of glycolysis and OXPHOS-related genes. This likely supports the high energy cell demand or the presence of an intermediate cell phenotype related to the concomitant presence of embryonic-like de-differentiated and functionally differentiated cardiomyocytes. Further analyses by single-cell RNA-seq would help to dissect the actual contribution of each population to this process.

Cell metabolism alterations are known to induce structural changes in glycans that modify proteins and lipids on the cell surface. Glycosylation alterations have been partially investigated in the heart[27,28] and in stem cell pluripotency and differentiation[55]. Only recently, the glycosylation of eight individual zebrafish organs including the heart was characterized, which revealed organ-specific glycosylation and sialylation[56], but this is the first time that glycosylation analysis has been done for the regenerating zebrafish heart. Specifically, we identified a disruption of membrane glycan structures and a remodeling to normal glycosylation taking place during the different phases of zebrafish heart regeneration.

We have previously reported the preferential binding of MAA, SNA-I and WGA lectins to blood vessels in rat hearts[27], suggesting a possible mechanism to regulate angiogenesis through sialylation of specific receptors such as vascular endothelial growth factor receptor-2 (VEGFR2), a pro-angiogenic receptor expressed by endothelial cells, whose α-(2,6)-sialylation has been demonstrated to play an important role in the modulation of VEGF/VEGFR2 interaction[57]. On the other hand, the sialylation in zebrafish hearts was not limited to blood vessels, but extended also to epicardial tissue. Altered expression of certain sialic acid types or their linkages is closely related to cellular adhesion and migration[58,59], which are required for zebrafish heart regeneration[60]. Hanzawa and collaborators have previously demonstrated that α-(2,6)-sialylation decreased during embryonic development from 6 to 48 h post-fertilization in zebrafish, suggesting a shift from embryonic α-(2,6)- to adulthood α-(2,3)-sialylation during development[61]. In our study, sialylation showed differences both in the degree and linkage type of sialylation. Specifically, α-(2,6)-sialylated N-linked structures were observed exclusively in low amounts in 2, 7 and 14 dpci tissues, and not in the sham, suggesting their relevance in the regeneration process, possibly through a shift to a more embryonic pattern.

Overall, our data collectively suggest a global downregulation of expressed sialylated O-glycan structures, and an upregulation of expressed sialylated N-glycan structures. This differential sialylation appears to be stage-specific, highlighting the significance of sialylation in regenerating zebrafish heart.

The lower expression of high mannose-type structures observed during the inflammatory phase could be a mechanism to induce inflammation resolution, as changes in mannose structures have been linked to the inflammatory processes[62]. High mannose structures have also been reported to be increased upon induction of dedifferentiation and acquisition of pluripotency[63]. In our study, we confirmed a dynamic high mannose profile during the regeneration process, showing a reduction in high mannose structures in the inflammatory phase, and a gradual increase in the following phases which may be linked to the dedifferentiation of cardiomyocytes.

Moreover, the presence of sialylated and sulfated O-linked core 1 structures in the injured tissue is consistent with previous studies demonstrating their relevant function in the context of the leukocyte recruitment. Circulating leukocytes constitutively express L-selectin, implicated in lymphocyte homing and recruitment of leukocytes at the site of inflammation[64,65], and sialylated and fucosylated[66], and/or sulfated[67,68] O-linked structures have been suggested to be specifically recognized by L-selectin.

In conclusion, we have shown distinctive metabolic alterations and membrane glycan remodeling during the transition through different phases of the zebrafish heart regeneration, and suggested that these processes are crucial within the regeneration process. In addition, to the best of our knowledge, this is the first structural analysis of N- and O-linked glycans in regenerating zebrafish heart. Ultimately, a further investigation of the underlying mechanisms which regulate the observed alterations could enable the identification of new therapies for the treatment of MI based on the re-induction of the regenerative capability.

## Methods

**Zebrafish lines and heart cryoinjury**. Zebrafish wild-type AB strain and transgenic strain cmlc2::GFP to analyze the injured area were obtained from the Zebrafish International Resource Centre and maintained under standard conditions at 28 °C[31,69]. Fish aged 6–18 months were anesthetized in 0.04% tricaine and placed ventral side up in a damp sponge. To induce heart damage, the cryoinjury method was used[36]. Briefly, a small incision was made through the chest with iridectomy scissors to access the heart, and the ventricular wall was frozen by applying for 25 s a stainless steel cryoprobe precooled in liquid nitrogen. The tip of the cryoprobe was 6 mm-long with a diameter of 0.8 mm. To stop the freezing of the heart, fish water at room temperature was dropped on the tip of the cryoprobe. Fish were swimming normally within half an hour. For analysis of regeneration, animals were euthanized at different times post-injury (2, 7 and 14 days) by immersion in 0.16% tricaine and hearts were dissected in media containing 2 U/ml heparin and 0.1 M potassium chloride (KCl). Control ventricles after sham operation consisted of healthy myocardium, which were not subjected to cryoinjury. All animal-related protocols were performed in accordance with National Guidelines and approved by the Animal Care Research Ethics Committee (ACREC) at the National University of Ireland, Galway, and the Health Product Regulatory Authority (HPRA), Ireland.

**RNA-sequencing**. Pooling of four ventricles was extracted in TRIzol™ (Thermo Fisher Scientific, USA) using a Qiagen TissueLyser set at 50 oscillations per min (Qiagen, USA) at 4 °C for 10 min. After precipitation by chloroform, RNA was purified using RNeasy® Mini Spin Columns (Qiagen, USA) and diluted in RNase-Free water (Qiagen, USA). RNA quality and concentration were determined using a bio-analyzer (Agilent Technologies, USA). After that, samples were sent to Genewiz (South Plainfield, NJ, https://www.genewiz.com) for RNA-sequencing. Briefly, after mRNA enrichment with Oligod(T) and fragmentation for 15 min at 94 °C, cDNA was generated with SuperScript II Reverse Transcriptase (Thermo Fisher Scientific, USA). Validation of the library was performed on the Agilent TapeStation (Agilent Technologies, USA), and quantification using Qubit 2.0 Fluorometer (Invitrogen, USA) as well as by quantitative PCR (Applied Biosystems, USA). After trimming, sequence reads shorter than 30 nucleotides were discarded. The remaining sequence reads were aligned to the *Homo sapiens* and *Danio rerio* reference genome with CLC Genomics Server program v. 9.0.1. Total gene hit counts and total transcript counts were measured and RPKM values were calculated.

After mapping and total gene hit counts calculation using CLC Genomics workbench v. 9.0.1, the results were used for downstream analysis of differential gene expression. Genes with adjusted $p$ value < 0.05 and absolute $\log_2FC > 1$ were considered differentially expressed.

**In situ sequencing**. Cryoinjured zebrafish heart tissue samples were fixed in PFA and embedded in OCT mounting medium, sectioned in 10-μm-thick cryosections, and stored at −80 °C until use. In situ sequencing (ISS), a targeted multiplexed mRNA detection assay employing padlock probes, rolling circle amplification (RCA), and barcode sequencing, was applied. Technical details and descriptions of the ISS method can be found published in Gyllborg et al.[70] Padlock probes targeting gfp, gapdh, pkma, mt-co1 and sdha were designed and tested. In short, sections were permeabilized with 100 ug/ml pepsin in 0.1 M HCl for 1 min and washed with PBS. mRNA was priming with random decamers, and reverse transcriptase (BLIRT, Poland) overnight (O/N) at 37 °C. Tissue sections were then fixed for 30 min with 3% PFA and subsequently washed with PBS. Phosphorylated padlock probes (PLPs) were hybridized at a final concentration of 10 nM/PLP and ligated with Tth Ligase (BLIRT). This was performed at 37 °C for 45 min and then moved to 45 °C for 1 h. Sections were washed with PBS and RCA was performed with phi29 polymerase (Monserate, USA) O/N at 30 °C. The sections were treated with TrueVIEW Autofluorescence Quenching kit (2BCScientific) for 30 s and immediately washed with PBS. Bridge-probes (10 nM) were hybridized at RT for 1 h in hybridization buffer (2× SSC, 20% formamide), followed by hybridization of readout detection probes (100 nM) and DAPI (Biotium, USA) in the hybridization buffer for 1 h at RT. The RCA products were decoded through three cycles of barcode sequencing imaging.

**Imaging setup and analysis**. Images were acquired using a Zeiss Axio Imager Z2 epifluorescence microscope (Zeiss, Germany), equipped with a ×20 objective. A series of images (10% overlap between two neighboring images) at different focal depths was obtained, and the stacks of images were merged into a single image thereafter using the maximum-intensity projection (MIP), and then stitched together in the Zeiss ZEN software. After exporting images in.tif format, images were aligned between cycles. The stitched images were then re-tiled into multiple smaller images, henceforth referred to as tiles. Tiled DAPI images were segmented with standard watershed segmentation. The tiled images with RCA products were

top-hat filtered, followed by initial nonlinear image registration, spot detection, fine image registration, crosstalk compensation, gene calling, and cell calling using the pipeline.

To identify the BZ, we took advantage of the absence of gfp expression in the injured area. From the point where gfp was still expressed (uninjured area), we considered the distance of 1 cm on a ×20 magnification. For the RZ an equivalent area in the region proximal to the bulbus arteriosus (on the opposite side of the ventricle with respect to the injury area), where there were no signs of damaged tissue, was selected.

In these two regions, each gene was quantified by dividing the spot total number for the number of gfp⁺ cardiomyocytes.

**Protein extraction for proteomic analysis**. Proteins were extracted from the pooling of 4 ventricles using RIPA buffer (Thermo Fisher Scientific, USA) containing cOmplete™ EDTA-free protease inhibitor cocktail (Roche, Switzerland). Homogenization was obtained through a Qiagen TissueLyser set at 50 oscillations per min (Qiagen, USA) for 15 min at 4 °C. After incubation on ice for 15 min and centrifugation at 12,000 rpm for 10 min at 4 °C, protein concentration was estimated using a Micro BCA™ Protein Assay Kit (Thermo Fisher Scientific, USA). Sample preparation for LC-MS/MS analysis was performed using a suspension trapping (S-Trap) method to concentrate samples and remove detergent interference. Specifically, an adapted digestion protocol based on S-trapTM Micro spin columns was used following the manufacturer's instructions. Tris-HCl was used in place of TEAB buffer (triethylammonium bicarbonate) both in SDS lysis buffer (1% SDS, 100 mM Tris-HCl, pH 7.55) and in binding buffer (90% aqueous methanol with a final concentration of 100 mM Tris-HCl, pH 7.1). Briefly, about 100 µg of estimated protein extracts was dried and resuspended in 25 µl of lysis buffer. Reduction of disulfide bonds was performed using 20 mM dithiothreitol (DTT) for 10 min at 95 °C, and the alkylation by the addition of iodoacetamide to a final concentration of 40 mM (30 min, dark). Undissolved matter was removed by centrifugation for 8 min at 13,000 g. Aqueous phosphoric acid at 1:10 was added (≈1.2% final concentration) and then 165 µl of binding buffer. The acidified SDS lysate/methanol buffer mixture was loaded into the micro-column for protein trapping and trypsinization (1:25-enzyme:protein) according to the manufacturer's protocol. After 1 h incubation at 37 °C, peptides were eluted using centrifugation at 4000 g with 40 µl each of 50 mM ammonium bicarbonate, then 0.2% formic acid (FA). Hydrophobic peptides were recovered with 35 µl of 50% acetonitrile containing 0.2% FA. Pooled elution of each sample was dried and resuspended in 25 µl of loading solution (2% acetonitrile, 0.1% trifluoroacetic acid). The peptide concentration was measured using Nanodrop™ One/Onec Microvolume UV-Vis Spectrophotometer (Thermo Fisher Scientific, USA).

**Proteomic analysis by nLC-ESI MS/MS label-free quantification**. Membrane proteins were quantified by BCA assay and submitted to bottom-up proteomic sample preparation for LC-MS/MS analysis. A suspension trapping (S-Trap) method was adopted to concentrate samples and to remove detergent interference[71]. About 1.5 µg of peptide mixtures was injected into the UHPLC system (Ultimate™ 3000 RSLCnano, Thermo Scientific, USA) coupled online with Impact HD™ UHR-QqToF (Bruker Daltonics, Germany). Each sample was analyzed at least twice to minimize technical variability. Samples were loaded onto a pre-column (Dionex, Acclaim PepMap 100 C18, cartridge, 300 µm) followed by a 50 cm nano-column (Dionex, ID 0.075 mm, Acclaim PepMap 100, C18). The separation was performed at 40 °C and at a flow rate of 300 nL/min using multistep 4 h gradients of acetonitrile as already reported[72]. The column was online interfaced to a nanoBoosterCaptiveSpray™ ESI source (Bruker Daltonics, Germany). Data-dependent-acquisition mode was applied based on CID fragmentation assisted by N2 as collision gas. Mass accuracy was improved using a specific lock mass (1221.9906 m/z) and a calibration segment (10 mM sodium formate cluster solution) before the beginning of the gradient for each single run. Acquisition parameters were set as already described[73].

Data elaboration was performed through DataAnalysis™ v.4.1 Sp4 (Bruker Daltonics, Germany) and protein identities and relative abundances were determined using Peaks Studio 8.5 (Bioinformatics Solutions Inc., USA)[74]. All duplicates for each sample were considered independently. An in-house constructed Uniprot's reference database of *Danio rerio* (accessed Feb 2018, 556,568 sequences; 199,530,821 residues) was combined with a decoy database and implemented in the software. For protein identification, the following parameters were specified: enzymatic digestion performed by trypsin, allowing one missed cleavage; precursor mass tolerance was 20 ppm; fragment mass tolerance of 0.05 Da; carbamidomethylation (Cys) as fixed modification. To determine the false-positive identification rate, the estimated spectra were used against the decoy database. An FDR of ≤1%, with a peptide score of −10 log p ≥ 20 was considered adequate for confident protein identification. De novo ALC score was set at ≥50%. Peptide feature-based quantification was performed to calculate the relative protein and peptide abundance in the samples. Only confidently identified peptide features were matched and used for the estimation of the related signal intensity. Likewise, the area under the curve of the extracted ion chromatograms was measured and used for the relative quantification between runs. Retention time shift tolerance was 6 min. Areas were normalized using total ion current. To get the summed cumulative peak area of the protein, only unique peptides that are assigned to

particular proteins were selected and only proteins that have more than two unique peptides were considered.

**Gene ontology enrichment and Ingenuity Pathway Analysis**. Gene ontology (GO) enrichment analysis was carried out using Database for Annotation, Visualization, and Integrated Discovery (DAVID, http://david.abcc.ncifcrf.gov). Functional classes of genes were identified by biological processes and biological processes GO terms were queried. Significant terms were identified as those with p < 0.05. The top ten biological processes for each time point were chosen after the removal of processes with greater than 80% redundancy of listed genes than that of a more statistically significant biological process. The R package GOplot was used for the generation of plots combining expression data with functional analysis. For Ingenuity Pathway Analysis (IPA®, Qiagen, USA, www.qiagen.com/ingenuity) gene symbols and orthologues were identified. An excel file containing the gene names, log₂ FC, p value and adjusted p value was created for each time point. The excel files were then combined according to Ensembl gene ID. The combined excel file was uploaded to the IPA® software applying cut-offs of log₂ FC > 1.5 and adjusted p < 0.05. Data from proteomic analysis were analyzed in a similar way, comparing the FC ratio of each time point to sham. The same cut-offs were applied.

**Membrane protein fraction extraction from zebrafish myocardial tissue**. Membrane protein fraction from injured and non-injured pooled ventricles was obtained using the Mem-PER™ Plus Membrane Protein Extraction Kit (Thermo Fisher Scientific, USA) following the manufacturer's protocol. In brief, the ventricles were washed in Cell Wash Solution and cut into small pieces using a scalpel. Next, permeabilization buffer containing cOmplete™ EDTA-free protease inhibitor cocktail (Roche, Switzerland) was added, and the suspension was incubated for 10 min at 4 °C. After centrifugation at 16,000 g for 15 min at 4 °C, the supernatant, corresponding to the cytosolic fraction, was discarded. The pellet, corresponding to the membrane fraction, was resuspended in the Solubilization buffer and incubated for 30 min at 4 °C with shaking. After centrifuging at 16,000 g for 15 min at 4 °C, the supernatant, containing the membrane proteins, was quantified using a Micro BCA™ Protein Assay Kit (Thermo Fisher Scientific, USA) according to the manufacturer's instructions and stored in aliquots at −80 °C until further use.

**Fluorescent labeling of membrane proteins**. Membrane proteins were diluted to 1 mg/ml in 100 mM sodium bicarbonate, pH 8.2, and all the following steps were carried out in the dark. Specifically, 1 mg of each sample was incubated with 50 µg of Alexa Fluor® 555 succinimidyl ester (Thermo Fisher Scientific, USA) dissolved in dry dimethyl sulfoxide. After incubation for 2 h at room temperature with gentle shaking (200 rpm), unincorporated dye was removed from the labeled samples using a 3 kDa molecular weight cutoff centrifugal filter with phosphate-buffered saline, pH 7.4 (PBS) elution. Labeled protein concentration and degree of label substitution were determined by measuring absorbances at 280 and 555 according to the manufacturer's instructions. An arbitrary E of 10 and molecular mass of 100 kDa were used for relative quantification[75].

**Lectin microarray analysis**. Lectin microarrays containing 49 immobilized lectins (Supplementary Table 5) were utilized to give specific information on the repertoire of glycans present in the sample[75]. To determine the optimal sample concentration for lectin microarrays, a titration from 1 to 10 µg/ml diluted in Tris-buffered saline supplemented with Ca²⁺ and Mg²⁺ (TBS; 20 mM Tris-HCl, 100 mM NaCl, 1 mM CaCl₂, 1 mM MgCl₂, pH 7.2) and 0.05% (v/v) Tween® 20 (TBS-T) of labeled proteins was carried out. The optimal concentration, corresponding to a non-saturated signal and low background (<300 relative fluorescence units), was 2.5 µg/ml. Samples were incubated on lectin microarrays for 1 h at room temperature[75]. After washing three times in TBS-T followed by one wash in TBS, and drying by centrifugation, microarray slides were scanned in a G2505 microarray scanner (Agilent Technologies, USA) using a 532 nm laser (5 µm resolution, 90% laser power). Data were saved as.tif files. Three biological replicates and three technical replicates were assessed.

Background-subtracted median fluorescence intensities were extracted from the.tif files using GenePix Pro v6.1.0.4 (Molecular Devices, UK). The median of six replicates was taken as one data point (per lectin) and technical replicates averaged per-array (i.e., per sample). Asialofetuin (ASF) was used as control glycoprotein and data were normalized to the per-subarray (i.e., per sample) total intensity mean across three replicate experiments. For binding data representation, dot plots were created using MATLAB® R2018a software (The MathWorks, Inc., USA, https://www.mathworks.com). Clustering analysis was conducted using Clustering Explorer v3.5 (HCE 3.5, University of Maryland, http://www.cs.umd.edu/hcil/hce/hce3.html) and Euclidean distance with complete linkage.

**Histochemistry**. Whole injured and non-injured hearts were explanted at each time point and fixed in 4% paraformaldehyde (PFA) in PBS O/N at 4 °C. After washing in PBS and infiltration in 30% sucrose solution O/N at 4 °C, they were embedded in optimal cutting temperature compound (OCT) medium (Sakura Finetek, USA). Embedded samples were stored at −80 °C and subsequently cut into 10 µm cryosections with a cryostat (Leica CM1850, Germany). Cryosections of whole hearts were collected on Superfrost Plus® slides (Thermo Fisher Scientific,

USA) and equally distributed onto six different slides so that each slide contained sections from all areas of the ventricle. For immunohistochemistry, samples were permeabilized with TBS-T (tris-buffered saline, Tween 20 0.05%), washed in TBS and then blocked with 10% donkey serum (Sigma-Aldrich) for 1 h at RT. The samples were incubated at 4 °C O/N with the following primary antibodies: rabbit polyclonal to CD163 (ab87099, Abcam, UK) at 1:200, mouse monoclonal to vimentin (RV202, Abcam, UK), rabbit polyclonal to Gata 4 (ab61170, Abcam, UK) at 1:100 and rabbit polyclonal to α-smooth muscle actin (ab5694, Abcam, UK) at 1:250. After washing with TBS-T, secondary antibodies conjugated to Alexa Fluor® 647 or Alexa Fluor® 555 (Jackson Laboratories, USA) were used at a dilution of 1:800. Mounting was performed with Prolong® Gold with DAPI (Thermo Fisher Scientific, USA).

For Alcian blue staining, sections were stained in Alcian blue (pH 2.5) for 40 min. After washing with running tap water, they were counterstained with nuclear fast red for 10 min. Next, sections were washed with running tap water and dehydrated with increasing concentrations of ethanol (50% for 10 s, 70% for 10 s, 90% for two min and absolute ethanol for 2 min).

For lectin histochemistry, slides were washed with TBS supplemented with 0.05% (v/v) Triton X-100 (TBS-T2). After blocking with 3% (w/v) periodate-treated bovine serum albumin (BSA) (Sigma-Aldrich) in TBS for 1 h, sections were washed and incubated with TRITC-conjugated lectins MAA, SNA-I and WGA (EY Laboratories, USA) in TBS-T O/N at 4 °C. Slides were then washed three times in TBS-T, once in TBS and mounted with Prolong® Gold with DAPI (Thermo Fisher Scientific, USA). To verify carbohydrate-mediated binding[76], lectins were also co-incubated with their respective haptenic sugars for 1 h before incubating the mixture on the tissue sections.

A laser confocal microscope (Olympus FluoView™ 1000, Olympus America) was used for all imaging and quantifications were performed using ImageJ software (Rasband, W.S., ImageJ, U. S. National Institutes of Health, USA).

**N- and O-glycans release from membrane proteins**. Membrane protein samples obtained from the pooling of 6 ventricles were applied to 30 kDa MWCO centrifugal filter (Millipore, USA). The filter was then washed with 7 M urea, 2 M thiourea and 40 mM Tris-HCl. After reduction with 25 mM dithiothreitol and alkylation with 62.5 mM iodoacetamide, the samples were digested with 1 μg (1% w/w) of trypsin (Promega, USA) and incubated at 37 °C O/N. Tryptic peptides were precipitated with 80% (v/v) acetone and air-dried. The pellet was washed twice in cold 60% methanol, resuspended in 50 μl of 50 mM ammonium bicarbonate (pH 8.4), and digested with 1 μl of PNGase F (Asparia Glycomics, Spain) at 37 °C O/N. N-glycans were separated from O-glycopeptides/polypeptides by SEP-Pak C18 cartridge (Waters Corporation, USA), which was pre-washed with methanol and conditioned with different dilutions (90% and 10%) of acetonitrile (ACN) in 0.5% trifluoroacetic acid (TFA). Released N-linked glycans were reduced by 0.5 M sodium borohydride (NaBH$_4$) and 20 mM NaOH at 50 °C O/N, while O-linked glycans were released by the β-elimination reaction. Samples were desalted with 25 μl of AG50WX8 cation-exchange resin (Bio-Rad, USA) and dried in a Savant SpeedVac[77].

**LC-ESI-MS/MS analysis**. Released glycans were resuspended in deionized water and analyzed by liquid chromatograph-electrospray ionization tandem mass spectrometry (LC-ESI/MS). The oligosaccharides were separated using a column (10 cm Å 250 μm) packed in-house with 5 μm porous graphite particles (Hyper-carb™, Thermo Fisher Scientific, USA). The oligosaccharides were injected into the column and eluted with an acetonitrile gradient (buffer A, 10 mM ammonium bicarbonate; buffer B, 10 mM ammonium bicarbonate in 80% acetonitrile). The gradient (0–45% buffer B) was eluted for 46 min, followed by a wash step with 100% buffer B, and equilibrated with buffer A in the subsequent 24 min. A 40 cm × 50 μm i.d. fused silica capillary was used as a transfer line to the ion source. The samples were analyzed in negative ion mode on an LTQ linear ion trap mass spectrometer (Thermo Electron, USA), with an IonMax™ standard ESI source equipped with a stainless steel needle kept at –3.5 kV. Compressed air was used as the nebulizer gas. The heated capillary was kept at 270 °C, and the capillary voltage was –50 kV. Full scan (m/z 380–2000, two microscan, maximum 100 ms, target value of 30,000) was performed, followed by data-dependent MS2 scans (two microscans, maximum 100 ms, target value of 10,000) with normalized collision energy of 35%, isolation window of 2.5 units, activation $q = 0.25$ and activation time 30 ms. The threshold for MS2 was set to 300 counts. Data acquisition and processing were conducted with Xcalibur™ software (Version 2.0.7, Thermo Fisher Scientific, USA). MS/MS spectra were used to identify glycan structures by manual annotation. The annotated structures were submitted to the unicarb-DR database (https://unicarb-dr.glycosmos.org/references/519) according to MIRAGE guidelines. The extracted ion chromatogram peak area was used to quantify individual structures. The area under the curve (AUC) of each structure was normalized to the total AUC and expressed as a percentage. The peak area was processed by Progenesis® QI (Nonlinear Dynamics Ltd., UK).

**Statistics and reproducibility**. All statistical analyses were conducted using Prism v6 (GraphPad, USA). Significance was assessed by one-way ANOVA with Tukey's post hoc correction for immunohistochemistry, lectin histochemistry and lectin microarray analyses. Differences were considered significant if $p < 0.05$. All the reported error bars indicate the standard deviation.

**Reporting summary**. Further information on research design is available in the Nature Portfolio Reporting Summary linked to this article.

## Data availability

All the annotated MS/MS spectra of identified N-glycan and O-glycan structures are available in unicarb-DR (https://unicarb-dr.glycosmos.org/references/519). Raw glycomic data files are available at https://glycopost.glycosmos.org/entry/GPST000053. The raw RNA-sequencing data generated during the current study are available in the Genome Sequence Archive (National Genomics Data Center, China National Center for Bioinformation, Beijing Institute of Genomics, Chinese Academy of Sciences) with the accession number GSA: CRA009013, and are publicly accessible at https://ngdc.cncb.ac.cn/gsa. All data of proteins identified with nano-LC-MS/MS are deposited in MassIVE and are available at https://massive.ucsd.edu/ProteoSAFe/dataset.jsp?task=661d66ff2c514775a3f052e550eaf498. All data that support the findings of this study are available from the corresponding author upon reasonable request.

## Code availability

The pipelines used for in situ sequencing analysis are available at https://github.com/Moldia. GOplots were created using the R package available via CRAN-The Comprehensive R Archive Network: http://cran.r-project.org/web/packages/GOplot.

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

## Acknowledgements

This work was funded by the European Commission funding under the AngioMatTrain 7th Framework Programme [Grant Agreement Number 317304, NMP-2014-646075, PITN-GA-2012-317304] and by the research grant from Science Foundation Ireland (SFI) co-funded under the European Regional Development Fund [Grant Number 13/RC/2073_P2]. The authors acknowledge the use of the facilities of the Centre for Microscopy and Imaging at the University of Galway, Galway, Ireland a facility that is co-funded by the Irish Government's Programme for Research in Third Level Institutions, Cycles 4 and 5, National Development Plan 2007–2013. We acknowledge the "In situ sequencing" unit at Science for Life Laboratory, Solna, Sweden, for technical assistance with the in situ sequencing experiments. The authors would like to thank Dr Raghvendra Bohara for editorial assistance, Dr Aileen Cronin and Dr Tracy Lynskey for help with maintenance of zebrafish lines, Dr Elke Rink for in vivo experiments design and Oliver Carroll for in vitro experimental design. This manuscript is dedicated in the memory of Anthony Sloan (CÚRAM).

## Author contributions

R.S., F.F. and P.C. designed the study, performed the experiments and wrote the article. A.A. performed lectin histochemistry. S.M-S. performed the sample processing and histology prior ISS analysis, and contributed to the revised version of the manuscript. C.J. and N.G.K. performed glycan structural analysis by LC-ESI-MS/MS. F.M. and C.C. performed proteomic analysis by nLC-ESI MS/MS. M.G. contributed to design in vivo experiments and directed the zebrafish facility. M.M.H. performed cell segmentation for ISS analysis. M.K. designed the lectin microarray experiment. A.P. designed the study and directed the project. All authors critically reviewed and approved the final manuscript.

## Competing interests

The authors declare no competing interests.
