## [Peer Review File · Communications Biology]

Reviewers' comments:

Reviewer #1 (Remarks to the Author):

In this manuscript, the authors present supporting evidence for the importance of metabolic reprogramming during zebrafish heart regeneration while also expanding about our knowledge of the potential of membrane glycan remodeling as an important component of the regeneration process. The authors use a combination of -omics approaches at different milestones of the regenerative process to uncover a metabolic switch and membrane glycosylation remodeling to possibly have vast implications in zebrafish heart regeneration.

Several papers have recently been published that describe the metabolic shift from mitochondrial oxidative phosphorylation to glycolysis in zebrafish heart regeneration and should be discussed in the introduction (eg Honkoop et al, 2020).

For RNAseq experiments, it is not clearly stated what samples were used in the differential expression analysis. Were all samples sham vs injured or across injured time points?

How do these RNAseq datasets compare to those that were previously published for cryoinjured hearts?

Can the authors clarify why they performed the experiments on whole ventricles as opposed to just the area surrounding the injury?

Besides stating that certain GO terms were enriched at each stage (lines 108-122), there's no discussion of what implications they might have each time point of regeneration and what information is usefully being pulled from these datasets.

In general, the first 3 figures are redundant and could be combined into one by highlighting only the most important analyses at each time point with more extended figures of the details at each timepoint in the supplemental data. It would also be interesting to add more direct comparative analyses between time points.

Previously it has been shown that most proliferative phase of regeneration is at 7dpi (Gonzalez-Rosa, 2011). Could the authors resolve why they see the peak proliferation phase at 14dpi?

Reviewer #2 (Remarks to the Author):

In this study the authors first performed bulk RNA-sequencing on whole zebrafish hearts isolated at different stages (2, 7 and 14 days) after cryoinjury and classify up- and downregulated genes. The report that genes with a role in energy metabolism are differentially expressed as well as genes involved in immune responses and fibrosis. Next, the authors describe a proteomics analysis to detect differences in protein abundance at these same stages. Finally, they investigate differences in membrane glycosylation at these stages.

My main concern is that this is a very descriptive study that does not go much beyond a description of the transcriptomics and proteomics data. The biological relevance of the data is limited as the study was done on whole heart preparations. Therefore, the work does not give new insights into the mechanisms of heart regeneration. The novelty of this manuscript is also limited as many transcriptomics studies on zebrafish heart regeneration have been published before (PMID: 32001677; PMID: 32877671; PMID: 31868166; PMID: 30597924; PMID: 29610343; PMID: 27241320; PMID: 25280539) as well as recent studies that give mechanistic insight into metabolic reprogramming during zebrafish heart regeneration (PMID: 32648304; PMID: 31868166). The data acquisitions seem technically well done, but there are issues with interpretation of the data.

Some specific comments:

1. In the introduction and throughout the manuscript the authors focus on cardiomyocyte proliferation. While this is an important process during heart regeneration, it is limited to a subset of cardiomyocytes that are located at the wound border zone. It is therefore surprising that the authors perform whole heart preparation for the RNA-seq and proteomics analysis as this will result in the inclusion of all cell types. With their conclusions on metabolic reprogramming, they again focus on cardiomyocytes and ignore all other cell types. This is relevant as metabolic reprogramming has been described in macrophages and endothelial cells. Without cell-type specific assays to address changes in energy metabolism the conclusions of the RNAseq and proteomics data are very limited and do not go beyond what is already known. No new mechanistic insight is provided into the regulation of heart regeneration.

2. The confocal images in the figures only show a zoom-in of the heart, Overview images would be needed to appreciate these images as it is now unclear where in the heart these images were taken. For example the authors relate the lectin binding to the epicardium, however without a clear overview picture one cannot see this. If indeed the epicardium is labelled by lectins, it might not be surprising that this is lost at 2 dpi as the epicardium is damaged and regenerated in 7-14 days. The epicardium is also easily damaged when removing the hearts from the fish, which might explain the biological variation.

Reviewer#1

1.1) Several papers have recently been published that describe the metabolic shift from mitochondrial oxidative phosphorylation to glycolysis in zebrafish heart regeneration and should be discussed in the introduction (eg Honkoop et al, 2020).

Response: We have discussed other papers recently published¹⁻³ following Reviewer' suggestion. We have added the following text to the manuscript' introduction (Page 3 and 4, lines 61-74):

An upregulation of glycolytic enzymes at the transcriptional level during cardiac regeneration has been reported in zebrafish¹. Moreover, a decrease in cardiomyocyte proliferation after injury was detected when glycolysis was impaired. Interestingly, the manipulation of pyruvate metabolism using Pyruvate dehydrogenase kinase 3 (Pdk3) and a catalytic subunit of the pyruvate dehydrogenase complex proved the pivotal role of glycolysis on cardiomyocyte proliferation after injury¹. Upregulation of genes that encode glycoenzymes and are regulators of pyruvate metabolism, such as Pdk2b, Pdk3b, and Pdk4 has been described in cell types surrounding damaged tissue^{1,2}. Indeed, the cardiomyocytes in the border zone seemed to undergo a metabolic shift from oxidative phosphorylation to glycolysis and lactate fermentation. Such a metabolic reprogramming to an embryonic-like metabolic state was regulated through Neuregulin1/Receptor protein-tyrosine kinase ErbB2 (Nrg1/ErbB2) pathway². Furthermore, the tumor suppressor gene Tumor protein 53 (Tp53), which has a role in cell cycle regulation, has been reported to be transiently suppressed by its negative regulator Mouse double minute 2 (Mdm2) during zebrafish heart regeneration³.

1.2) For RNAseq experiments, it is not clearly stated what samples were used in the differential expression analysis. Were all samples sham vs injured or across injured time points?

Response: We regret the lack of clarity regarding this matter. The differential expression analysis of RNAseq experiments were all sham vs injured samples. The following sentence was added to the manuscript to clarify this point (Page 5, lines 117-118), and it was also specified in Figure 1:

Sham vs injured samples were compared to define differentially expressed genes (DEG) (Page5, lines 115-116).

1.3) How do these RNAseq datasets compare to those that were previously published for cryoinjured hearts?

Response: We have compared our results with the previously reported RNAseq datasets. Due to the wide variability among the aim, experimental design (including timepoints studied), and fish strains in the papers available in the literature, we have prepared a table resuming the main results and compared them with our findings. (Supplementary Table S7)

Table S7: Literature summary of the most recent publications describing RNAseq datasets in zebrafish' cryoinjury model and comparison with our results.

Aim	Model and/or fish strain	Early post-injury stage (from 0 to 6dpi)	Mid-term post injury (from 7dpi to 13pi)	Long-term remodelling (from 14 dpi)	Comparison with our results	Ref
Computational analysis of published ablation model datasets to select 15 candidate genes with the potential role in heart regeneration; Comparison with a cryoinjured dataset at 5 timepoints: control, 1, 3, 5 and 7 days after operation	WIK wild-type fish	NA	NA	NA	Limited amount of data since only 15 candidate genes were analysed. The computational comparison showed concordance across two different models at 1 and 3 dpi, but there is no comparison between time points.	2014 ⁴
Generation and integration of a co-expression network of heart regeneration in zebrafish.	Wild type	Massive cell death from 4hpi; Lack of cardiomyocytes and fibrotic scar formation at 3dpi; Enrichment of genes implicated in energy metabolism, amino-acid biosynthesis and DNA replication, which may suggest cell proliferation,	Boost of regeneration at 7 dpi (enrichment in peptidase activity, DNA metabolism and replication enrichment)	No differences can be detected at 90 dpi with controls since remodelling is complete	As in our group, the largest numbers of differentially-expressed genes were detected at the early stages of regeneration. Authors performed a longer time-depended assessment at 4 hours (hpi), 1, 3, 7, 14 and 90 dpi.	2016 ⁵

Study of collagen-producing cells subpopulations and their fate after zebrafish myocardium regeneration using different transgenic strains that present a genetic ablation of collagen producing cells	Tg(postnb:citrine)^{cn6}; NA Tg(periostin:CreER^{T2})^c ⁿ⁷; Tg(col1a2:loxPtagBFP -loxP-mCherr- NTR)^{cn8}; Tg(col1a2:loxP- mCherry-NTR)^{cn11}; Tg(fli1a:CreERT2)^{cn9}; Tg(kdrl:mcherry) ; Tg(-3.5ubb:loxP-loxP- eGFP)^{cz1702Tg} ; Tg(ubb:Switch)^{cz1701};T g(ubb:mCherry)^{cz1705T} ^g ; Tg(fli1a:GFP)^{Y1}, Tg1(- 6.8wt1a:EGFP)^{li7Tg}; Tg(tcf21:CreERT2)^{pd42} Tg,	NA	ECM genes upregulated at 7dpi	Decrease in ECM gene expression at 60 dpi there were also collagen- encoding genes that were up- regulated at 60 dpi	Focused on ECM production and accumulation during zebrafish remodelling after cryoinjury. Study of gene expression at day 7 and 60 dpi, and comparison with the uninjured expression. At 7dpi the upregulated pathways such as “Hepatic Fibrosis/Hepatic Stellate Cell Activation”, “Leukocyte Extravasation Signalling” or “Axonal Guidance Signalling”, are consistent with our data. However, authors sorted cell populations before the RNAseq analysis.	2018⁶
Dynamic transcriptome response monitoring of mRNA and microRNA in zebrafish at 1–160 dpi	Adult wild type	Increase of regeneration at 1 and 4dpi	Largest peaks of differentially expressed genes compared to controls were found at 4 and 7dpi.	From 14dpi an enrichment in DNA packaging, ECM organization and cell adhesion reported;	Focused on identifying miRNA-mRNA interactions to describe potential therapeutic targets for MI. Spatio-temporal resolution of the mRNAs during heart regeneration at different stages after injury. Similar results related to cell proliferation enrichment during the first 2 weeks after the injury. No linkage assessed within the metabolic stage and glycosylation during regeneration.	2018⁷

				Using GO gene sets, and enrichment in proliferation, immune response and migration were reported.	From 60dpi authors reported an enrichment in cardiovascular development, heart contraction and cell growth	
Single-cell RNA-seq analysis to explore the differential mechanisms between non-proliferating and proliferating cardiomyocytes	Tupfel Long Fin (TL); Tg(myl7:dsRED) ^{s879Tg} ; Tg(myl7:GFP) ^{twu34Tg} ; Tg(phd3:GFP) ^{sh144} ; Tg(gata4:EGFP) ^{ae1} ; TgBAC(nppa:mCitrine)Tg(cmhc2:CreER) ^{pd10} ; Tg(β act2:BSNrg1) ^{pd107} hearts; Tg(gata4:EGFP) ^{ae1} hearts; Tg(phd3:GFP) ^{sh144} hearts;	NA	NA	NA	Consistency with our results related to a metabolic shift in adult cardiomyocytes from OXPHOS to glycolysis during heart regeneration. From a mechanistic point of view, authors confirmed their pivotal role in metabolic reprogramming by blocking Nrg1/ErbB2 signaling. No differential analysis and comparison performed over different timepoints after cryoinjury.	2019 ²
Study of cardiomyocyte proliferation after injury in adult zebrafish through metabolic reprogramming from a	Tg(cryaa:DsRed ,- 5.1 myl7:CreERT2) ^{pd10} ; pkma2 ^{s717} ; pkmb ^{s718} ; and ppargc1a ^{bns176} ;	NA	NA	NA	Mechanistic analysis of glycolysis by Pdk3 seems to promote cardiomyocyte de-differentiation after injury. However, only on timepoint at 5dpi was used.	2020 ¹

mechanistic point of view by the manipulation of pyruvate metabolism using Pdk3 as well as a catalytic subunit of the pyruvate dehydrogenase complex (PDC)	Tg(cryaa:Cerulean,hs p70l:LOXP-STOP-LOXP-pdha1aSTA-T2A-mCherry)^{bns343}; Tg(cryaa:Cerulean,hs p70l:LOXP-STOP-LOXP-pdk3b-T2A-mCherry)^{bns344}				Consistency in the results about the metabolic shift from OXPHOS to glycolysis. Authors reported a reduction in mitochondrial gene expression that indicates lower mitochondrial activity, thereby reducing OXPHOS during regeneration.	
Study of the transcriptome during heart regeneration, with a focus on the role of the suppression of transcription factor Tp53 associated genes.	Wild-type or transgenic zebrafish of the EK/AB strain β-actin2:loxp-mCherry-STOP-loxp-DTA^{pd36}; β-actin2:loxp-mTagBFP-STOP-loxp-nrg1^{pd107}; β-actin2:loxp-mTagBFP-STOP-loxp-vegfaa^{pd262}; cmlc2:CreER^{pd10}; gata4:EGFP ; gata4:dsRed2^{pd28} ; tp53^{M214K}	NA	A high index of cardiomyocyte proliferation at 7dpi, Downstream of the Tp53 transcriptional network at 7dpi	A high index of cardiomyocyte proliferation at 14dpi	As reported in our work, the upstream regulation of factors involved in inflammation, such as TNF-α and IL-6. Results of earlier timepoints after injury not available. Mechanistic asses of Tp53 role in cardiomyocyte differentiation and division after cryoinjury by its suppression by mdm2	2020 ³

Study of macrophage response in zebrafish, as well as a comparison between neonatal and adult conditions of heart repair in mice proving their role in fibrosis and scar formation in the injured heart.	Wild type, TgBAC(mpeg1:βirA-Citrine)^{ox122} ; Tg(βactin:Avi-Cerulean-RanGap)^{ct700a} ; TgBAC(mpeg1:βirA-Citrine;βactin:Avi-Cerulean-Rangap)^{ox133} ; Gt(foxd3Citrine)^{ct110} ; Tg(mpeg1:EGFP)^{gl22} ; Tg(mpeg1:mCherry)^{gl}	Increased macrophage infiltration at 1dpi; genes related to inflammatory response upregulated	From 5dpi upregulation of genes related to regeneration	Resolution of inflammation by 14dpi	The experimental design focused on the study of macrophage transcriptome at 1dpi, 5dpi and 14dpi, and the comparison with the resection model. Authors proved the role of macrophages in the scar formation and remodelling.
--	---	---	---	-------------------------------------	--

1.4) Can the authors clarify why they performed the experiments on whole ventricles instead of just the area surrounding the injury?

Response: We thank the Reviewer for pointing out this experimental aspect. The border zone (BZ) myocardium, which surrounds the damage, has recently gained increased interest due to its probable significance in postinjury cardiac regeneration in zebrafish. For example, proliferating cardiomyocytes have been mainly detected in the zone bordering the injury⁹ and de-differentiation of cardiomyocytes has been confirmed in the BZ².

However, several lines of evidence suggest that the whole ventricle is involved in the regeneration process. Indeed, endothelial cell proliferation was detected not only in the injury area, but also in the periphery, implying the existence of paracrine signals¹⁰. Also, an increase in the number of autophagic vesicles has been evidenced both in regions proximal and distal to the injured area¹¹. Initially, our study aimed not to compare injured vs non-injured area, but “healthy” vs injured ventricles, including the deep changes occurring even far from the injury zone, so we decided to perform the experiments on whole ventricles.

For these reasons, we decided to start our investigation from a global transcriptomic profile of the whole ventricle to obtain a high sequencing depth which permitted to highlight and further investigate the relevance of metabolic reprogramming and glycan remodeling in regenerating zebrafish heart. However, in the revised version of the manuscript, we have addressed changes in energy metabolism in cardiomyocytes using targeted in situ single cell analysis and compared border zone (BZ) and remote zone (RZ)). The results evidenced a metabolic reprogramming from OXPHOS to glycolysis in BZ cardiomyocytes at 7 dpci, confirming previous studies². Moreover, we identified a simultaneous induction of glycolysis and OXPHOS related genes at 14 dpci in cardiomyocytes from both BZ and RZ. (See response **2.1** and Results page 9-10, lines 206-230)

1.5) Besides stating that certain GO terms were enriched at each stage (lines 108-122), there’s no discussion of what implications they might have each time point of regeneration and what information is usefully being pulled from these datasets.

Response: As suggested by the Reviewer, we have updated the discussion section (Page 16-17, lines 364-376) adding the following text:

We first investigated the transcriptional changes occurring during zebrafish heart regeneration using a deep sequencing approach. Enrichment analysis of DEG revealed time point-specific responses,

identifying GO terms related with immune processes, such as “neutrophil chemotaxis” and “inflammatory response”, as the most enriched at 2 dpci and 7 dpci, respectively.

Previous studies have demonstrated that the immune response plays a critical role in initiating tissue regeneration^{8,12–14}, and our results are in agreement with these findings, confirming the relevance of the immune processes in the early phases of regeneration. GO terms related to protein synthesis were among the most enriched at 14 dpci, indicating that the cells are prepared to create a new set of proteins that, when combined with extensive protein degradation, will allow the cell to undertake the substantial cellular proteome alterations that reprogramming necessitates¹⁵.

Interestingly, the GO terms “carbohydrate metabolic process” and “protein N-linked glycosylation” were enriched in all the time points analysed, suggesting the importance of metabolic regulation and glycan remodeling during the regeneration process.

1.6) In general, the first 3 figures are redundant and could be combined into one by highlighting only the most important analyses at each time point with more extended figures of the details at each timepoint in the supplemental data. It would also be interesting to add more direct comparative analyses between time points.

Response: As suggested by the Reviewer, we have combined the first 3 figures into one (Figure 1), showing only the most relevant analyses at each time point. The remaining analyses were included in the Supplementary Figure S4, in which we have also added direct comparative analyses between time points for RNA sequencing and protein MS experiments (Supplementary Figure S4j and i).

Sham vs 2 dpci

Sham vs 7 dpci

Sham vs 14 dpci

Fig. 1. The response to injury is characterized by molecular and metabolic changes.

a GOCircle plot summarizing gene ontology enrichment analysis at 2 dpci, **e** 7 dpci and **i** 14 dpci; “carbohydrate metabolic process” and “N-linked glycosylation” are identified as significantly activated processes, among others. The significance of each term ($-\log_{10}$ adjusted p -value) is specified by the height of the bar plot in the inner ring, while the color corresponds to the z-score. The expression level (\log_2FC) for the genes in each term is displayed in the outer ring scatterplots; red dots indicated up-regulated genes and blue dots down-regulated genes. **b** The most statistically significant canonical pathways for the transcriptomic analysis at 2 dpci, **f** 7 dpci and **j** 14 dpci identified using the IPA® software are listed according to their p -value ($-\log$); the threshold $-\log(p\text{-value}) = 1.3$ corresponds to a p -value of 0.05. Activation is indicated in red, inhibition in green and unpredictability in grey. **c** The most statistically significant canonical pathways for the proteomic analysis at 2 dpci, **g** 7 dpci and **k** 14 dpci. **d** Heatmap showing glycolytic enzymes and mitochondrial oxidative phosphorylation proteins at 2 dpci, **h** 7 dpci and **l** 14 dpci. Two biological replicates deriving from the pooling of 4 different animals ($n = 8$ animals per group) were analysed for all experiments.

Fig. S4. Upstream regulators at the different phases of regeneration. **a** Upstream regulators at 2 dpi obtained analysing the RNA-sequencing data and using the IPA® URA tool. The most significant cytokine and growth factors are shown according to their $-\log(p\text{-value})$ and z-score. **b** Confocal images and **c** relative quantification showing CD163 expression in the Sham and following injury. Cardiomyocytes are identified by GFP positivity. **d** Upstream regulators at 7 dpi identified as described in **a**). **e** Confocal images and **f** relative quantification showing Vimentin expression in the Sham and following injury. **g** Upstream regulators at 14 dpi identified as described in **a**). **h** Confocal images and **i** relative quantification showing Gata4 expression in the Sham and following injury. Scale bar = 10 μm . (* $p < 0.05$, ** $p < 0.01$, *** $p < 0.001$, **** $p < 0.0001$). Data are shown as mean with SD (n = 4 animals per group). **j** Comparison analysis between the different time points of the most significant pathways obtained analysing RNA-sequencing data. **k** inference of cell type-proportions (RNAseq deconvolution) obtained using the CellMapper package **l** Comparison analysis between the different time points of the most significant pathways identified using MS data. Two biological replicates deriving from the pooling of 4 different animals (n = 8 animals per group) were analysed for RNA-sequencing and MS experiments.

1.7) Previously it has been shown that most proliferative phase of regeneration is at 7dpi (Gonzalez-Rosa, 2011). Could the authors resolve why they see the peak proliferation phase at 14dpi?

Response: as Reviewer pointed out, our data shows a peak of proliferation at 14 dpi in contrast with the previously reported data by Gonzalez-Rosa that showed that peak at 7dpi. However, these results are not necessarily contradictory. Our results showed a clear increase in proliferation from 7dpi, reaching its peak at 14 dpi. Experimental design in Gonzalez-Rosa lacks an intermediate time point between 7dpi and 21dpi. Also, since a remarkable decrease in cell proliferation from 21dpi is widely described, we can easily assume that the peak of proliferation could be reached between 7dpi and 21dpi when post-injury cell debris is cleared and fibrotic tissue is replaced by cardiac tissue¹⁰. Indeed, according to Klett et al.⁷, cell proliferation is still up-regulated during the phase from 14dpi to 45dpi, showing a progressive increase of processes related to ECM remodelling which become dominant in the late phase.

Reviewer#2

2.1) In the introduction and throughout the manuscript the authors focus on cardiomyocyte proliferation. While this is an important process during heart regeneration, it is limited to a subset of cardiomyocytes that are located at the wound border zone. It is therefore surprising that the authors perform whole heart preparation for the RNA-seq and proteomics analysis as this will result in the inclusion of all cell types. With their conclusions on metabolic reprogramming, they again focus on cardiomyocytes and ignore all other types. This is relevant as metabolic reprogramming has been described in macrophages and endothelial cells. Without cell-type specific assays to address changes in energy metabolism the conclusions of the RNAseq and proteomics data are very limited and do not go beyond what is already known. No new mechanistic insight is provided into the regulation of heart regeneration.

Response: We thank the Reviewer for His/Her thoughtful comment. As it was noted, RNAseq and proteomics analysis performed on whole ventricles implicate the inclusion of all cell types, but, on the other side, enabled a high sequencing depth. To partially address the sample heterogeneity issue, we applied a computational method to infer cell type-specific expression information directly from bulk RNA-seq results, learning about cell-type proportions (RNAseq deconvolution). In particular, we employed the CellMapper package, available on the Bioconductor. This package infers cell type-specific expression based on co-expression similarity with known cell type marker genes. As expected, the results indicated that the cell type with the greatest number of significantly expressed genes were cells specific for the immune system, confirming the previous pathway analysis results, followed by cardiomyocytes.

From Fig. S4. Upstream regulators at the different phases of regeneration. Inference of cell type-proportions (RNAseq deconvolution) obtained using the CellMapper package.

Moreover, as the Reviewer suggested, we investigated changes in energy metabolism in a specific cell type. Based on the previous computational analysis of bulk RNAseq, we decided to address energy metabolism changes in cardiomyocytes using targeted in situ sequencing, comparing BZ (border zone) and RZ (remote zone) cardiomyocytes. In particular, the analysis was performed on the transgenic zebrafish line cardiac myosin light chain 2 (*cm1c2*), expressing the green fluorescent protein (gfp) under the control of *cm1c2* promoter, so GFP was chosen as a cardiomyocyte marker. The expression of glyceraldehyde-3-phosphate dehydrogenase (*gapdh*), pyruvate kinase M1/2a (*pkma*), mitochondrially encoded cytochrome c oxidase I (*mt-co1*) and succinate dehydrogenase complex flavoprotein subunit A (*sdha*) was investigated at a single-cell level and compared between cardiomyocytes of the border zone (BZ) and those of the remote zone (RZ). The results evidenced a metabolic reprogramming from OXPHOS to glycolysis in BZ cardiomyocytes at 7 dpci, indicating the activation of an embryonic phenotype due to de-differentiation processes as previously demonstrated². Moreover, we identified a simultaneous induction of glycolysis and OXPHOS related genes at 14 dpci in cardiomyocytes from both BZ and RZ, which is due to an increased energetic need and/or the presence of an intermediate phenotype related to the concomitant presence of embryonic-like de-differentiated and functionally differentiated cardiomyocytes.

Fig. 2. Cardiomyocyte targeted in situ sequencing. A targeted multiplexed mRNA detection assay employing padlock probes, rolling circle amplification (RCA), and barcode sequencing, was applied to detect spatial distribution of **a** gapdh **b** pkma **c** mt-co1 and **d** sdha at 2 dpi, 7 dpi and 14 dpi. All the experiments were performed on *cmlc2* zebrafish transgenic strain, where a green fluorescent protein (*gfp*) gene was inserted under the promoter of cardiomyocyte-specific gene cardiac myosin light chain 2 (*cmlc2*). For this reason, *gfp* was chosen as a cardiomyocyte marker. IA: injured area; BZ: border zone; RZ: remote zone. **e** Heatmaps showing the expression comparison of gapdh, pkma, mt-co1 and sdha between BZ and RZ at each time point. Heat legend represents relative quantification of spot number. Two biological replicates were analyzed.

2.2) The confocal images in the figures only show a zoom-in of the heart. Overview images would be needed to appreciate these images as it is now unclear where in the heart these images were taken. For example the authors relate the lectin binding to the epicardium, however without a clear overview picture one cannot see this. If indeed the epicardium is labelled by lectins, it might not be surprising that this is lost at 2 dpi as the epicardium is damaged and regenerated in 7-14 days. The epicardium is also easily damaged when removing the hearts from the fish, which might explain the biological variation.

Response: As suggested by the Reviewer, we have replaced images in Figure 4. Now the injured area is clearly identified by dashed lines and the GFP absence.

The epicardium in the injured area is, of course, damaged during the cryoinjury procedure. Having been aware that the epicardium contributes essential cells and signals during regeneration, we used extra care when removing the hearts. Specifically, to be sure that the epicardium would not have been damaged while explanted, we paid particular attention during the procedure of removing the heart from the chest, holding it by the bulbus arteriosus.

Fig. 4. Sialic acid distribution in regenerating zebrafish heart identified by lectin histochemistry.

a MAA, recognizing α -(2,3)-linked sialic acid, binding in ventricle tissue, and **b** MAA binding intensity relative quantification. **c** SNA-I binding to α -(2,6)-linked sialic acid within the ventricle tissue and **d** SNA-I binding intensity relative quantification. **e** WGA binding to sialic acid and GlcNAc residues in the ventricle tissue, and **f** WGA binding intensity relative quantification. The lectin binding quantifications highlighted a similar decrease at 2 dpci and a subsequent increase in the following time points. Cardiomyocytes were identified by GFP positivity. IA: injured area. Data are shown as mean with SD (n = 4 animals per group). Scale bar = 10 μ m. (* p < 0.05, ** p < 0.01, *** p < 0.001, **** p < 0.0001).

References

1. Fukuda, R. *et al.* Stimulation of glycolysis promotes cardiomyocyte proliferation after injury in adult zebrafish. *EMBO Rep* **21**, e49752 (2020).
2. Honkoop, H. *et al.* Single-cell analysis uncovers that metabolic reprogramming by ErbB2 signaling is essential for cardiomyocyte proliferation in the regenerating heart. *Elife* **8**, e50163 (2019).
3. Shoffner, A., Cigliola, V., Lee, N., Ou, J. & Poss, K. D. Tp53 Suppression promotes cardiomyocyte proliferation during zebrafish heart regeneration. *Cell Rep* **32**, 108089 (2020).
4. Rodius, S. *et al.* Transcriptional response to cardiac injury in the zebrafish: systematic identification of genes with highly concordant activity across in vivo models. *BMC Genomics* **15**, 852 (2014).
5. Rodius, S. *et al.* Analysis of the dynamic co-expression network of heart regeneration in the zebrafish. *Sci Rep* **6**, 26822 (2016).
6. Sánchez-Iranzo, H. *et al.* Transient fibrosis resolves via fibroblast inactivation in the regenerating zebrafish heart. *Proc Natl Acad Sci U S A* **115**, 4188–4193 (2018).
7. Klett, H. *et al.* Delineating the dynamic transcriptome response of mRNA and microRNA during zebrafish heart regeneration. *Biomolecules* **9**, 11 (2018).
8. Simões, F. C. *et al.* Macrophages directly contribute collagen to scar formation during zebrafish heart regeneration and mouse heart repair. *Nat Commun* **11**, 600 (2020).
9. Wu, C.-C. *et al.* Spatially resolved genome-wide transcriptional profiling Identifies BMP signaling as essential regulator of zebrafish cardiomyocyte regeneration. *Dev Cell* **36**, 36–49 (2016).
10. González-Rosa, J. M., Martín, V., Peralta, M., Torres, M. & Mercader, N. Extensive scar formation and regression during heart regeneration after cryoinjury in zebrafish. *Development* **138**, 1663–1674 (2011).
11. Chávez, M. N. *et al.* Autophagy Activation in zebrafish heart regeneration. *Sci Rep* **10**, 2191 (2020).
12. Bevan, L. *et al.* Specific macrophage populations promote both cardiac scar deposition and subsequent resolution in adult zebrafish. *Cardiovasc Res* **116**, 1357–1371 (2020).
13. Huang, W.-C. *et al.* Treatment of glucocorticoids inhibited early immune responses and impaired cardiac repair in adult zebrafish. *PLoS One* **8**, e66613 (2013).
14. de Preux Charles, A.-S., Bise, T., Baier, F., Marro, J. & Jaźwińska, A. Distinct effects of inflammation on preconditioning and regeneration of the adult zebrafish heart. *Open Biol* **6**, 160102 (2016).
15. Mizushima, N. & Klionsky, D. J. Protein turnover via autophagy: implications for metabolism. *Annu Rev Nutr* **27**, 19–40 (2007).

Reviewers' comments:

Reviewer #2 (Remarks to the Author):

The authors have tried to address my main concern about the cell type specificity of their RNAseq data. The used method of RNA in situ hybridization is a good way to demonstrate cell specificity, but I have a number of concerns about the interpretation of their results. This may be resolved by providing better quality images and zooming in on the regions of interest (see below). Besides this I still have a number of concerns which are addressed below point-by-point.

Specific points:

The authors provide a new figure Fig2 to address which cell type may be responsible for the changes in metabolic gene expression and where that cell type is located. From this they conclude that the differential expression of glycolytic genes and of mitochondrial genes observed in the RNAseq data is mainly due to different expression levels between BZ and RZ cardiomyocytes. It is unclear how the authors reach these conclusions as the claimed differences between BZ and RZ are not visible. First of all, a sham control heart is missing here. Second, only whole heart images are shown in which one can see very large spots and it is unclear what these are and in which cell type these are located. The authors have used a *myl7:gfp* transgene to mark CMs but this is not visible at this low magnification. In addition to these whole heart overviews the authors should show representative images with higher magnification and high resolution of the BZ and RZ in which it is possible to distinguish different cell types. Possibly they need to separate the channels for better visualization. In addition, the authors should clearly explain how the quantification was performed of which the results are shown in the heatmaps. For example, how is the quantification performed on GFP+ CMs and what is their definition of the BZ

figS4: Images shown are of too low resolution. Also the labeling of these images is unclear due to the small font size and low resolution. It is also unclear where the images are taken. Whole heart images should be provided along with a box around the area that is shown in the magnification.

Fig.4 and page 13: Authors claim that the lectin histochemistry mainly labels blood vessels. Based on the low-res pictures shown this seems difficult to claim. It also seems that WGA is labeling some cardiomyocytes in the BZ, which is ignored. Authors should substantiate their claim with an endothelial co-labeling or change the wording to tone down their conclusion about the specific cell type.

For all figures: all images of injured fish heart should be shown in a comparable manner in order to make the comparison easier. Images of whole hearts and sections with immunohistochemistry staining should be placed with the apex pointing down and the bulbus up.

Point-by-point response to Reviewer 's comments:

Specific points:

1. The authors provide a new figure Fig2 to address which cell type may be responsible for the changes in metabolic gene expression and where that cell type is located. From this they conclude that the differential expression of glycolytic genes and of mitochondrial genes observed in the RNAseq data is mainly due to different expression levels between BZ and RZ cardiomyocytes. It is unclear how the authors reach these conclusions as the claimed differences between BZ and RZ are not visible. First of all, a sham control heart is missing here. Second, only whole heart images are shown in which one can see very large spots and it is unclear what these are and in which cell type these are located. The authors have used a *myl7:gfp* transgene to mark CMs but this is not visible at this low magnification. In addition to these whole heart overviews the authors should show representative images with higher magnification and high resolution of the BZ and RZ in which it is possible to distinguish different cell types. Possibly they need to separate the channels for better visualization. In addition, the authors should clearly explain how the quantification was performed of which the results are shown in the heatmaps. For example, how is the quantification performed on GFP+ CMs and what is their definition of the BZ

Response: We thank the Reviewer for acknowledging the relevance of the new revised Fig.2. The purpose of our analysis was not to compare the injured heart with the sham but to focus on the BZ and RZ of each time point. However, we agree that including the sham control in this figure could be helpful for a visual evaluation of the expression in control. We have implemented this in Fig.2. Moreover, as requested, representative images having both higher magnification and better resolution of BZ and RZ at the different time points were added in a new supplementary figure (Fig. S6), in which channels were separated to facilitate the readers' comprehension.

Also, we added a paragraph in which BZ and RZ are defined, and we explain how the quantification was achieved now included in the Material and Methods (page 23, lines 518-524):

*To identify the BZ, we took advantage of the absence of *gfp* expression in the injured area. From the point where *gfp* was still expressed (uninjured area), we considered the distance of 1 cm on a 20X magnification. For the RZ an equivalent area in the region proximal to the bulbus arteriosus (on the opposite site of the ventricle with respect to the injury area), where there were no signs of damaged tissue, was selected.*

*In these two regions, each gene was quantified by dividing the spot total number for the number of *gfp*⁺ cardiomyocytes.*

Fig. 2. Cardiomyocyte targeted in situ sequencing. A targeted multiplexed mRNA detection assay employing padlock probes, rolling circle amplification (RCA), and barcode sequencing

was applied to detect the spatial distribution of **a** gapdh **b** pkma **c** mt-co1 and **d** sdha in sham and at 2 dpci, 7 dpci and 14 dpci. All the experiments were performed on cmlc2 zebrafish transgenic strain, where a green fluorescent protein (*gfp*) gene was inserted under the promoter of cardiomyocyte-specific gene cardiac myosin light chain 2. For this reason, *gfp* was chosen as a cardiomyocyte marker. IA: injured area; BZ: border zone; RZ: remote zone. Scale bar = 100 μ m **e** Heatmaps showing the expression comparison of gapdh, pkma, mt-co1 and sdha between BZ and RZ at each time point. Heat legend represents the relative quantification of the spot number. Two biological replicates were analyzed.

Fig. S6. Cardiomyocyte targeted in situ sequencing. Representative zoom in images of targeted multiplexed mRNA detection assay showing spatial distribution of spots identifying *gapdh*, *pkma*, *mt-co1* and *sdha* expression at 2, 7 and 14 dpci in the border and remote zones. A transgenic zebrafish line, in which a green fluorescent protein (*gfp*) gene was inserted under

the promoter of cardiomyocyte-specific gene cardiac myosin light chain 2 (*cm1c2*), was employed to identify cardiomyocytes. IA: injured area; BZ: border zone; RZ: remote zone. Scale bar = 100 μ m. Two biological replicates for each group were analysed.

2. figS4: Images shown are of too low resolution. Also the labeling of these images is unclear due to the small font size and low resolution. It is also unclear where the images are taken. Whole heart images should be provided along with a box around the area that is shown in the magnification.

Response: We thank the Reviewer for pointing out this issue with images resolution. We have now provided new Fig. S4 and S5 in which: i) the resolution is higher; ii) labelling size has been increased; and iii) heart images are shown at a lower magnification with the injured area clearly identified. Also, a representative selected area is displayed at a higher magnification. Specifically, in the Gata 4 immunostaining (Fig. S4h), since the aim was to show and quantify intranuclear staining, the images were kept at a higher magnification.

Fig. S4. Upstream regulators at the different phases of regeneration. **a** Upstream regulators at 2 dpci obtained analysing the RNA-sequencing data and using the IPA® URA tool. The most significant cytokine and growth factors are shown according to their $-\log(p\text{-value})$ and z-score. **b** Confocal images and **c** relative quantification showing CD163 expression in the Sham and following injury. Cardiomyocytes are identified by GFP positivity. Scale bar = 100 μm . **d** Upstream regulators at 7 dpci identified as described in **a**. **e** Confocal images and **f** relative quantification showing vimentin expression in the Sham and following injury. Scale bar = 100 μm . **g** Upstream regulators at 14 dpci identified as described in **a**. **h** Confocal images and **i** relative quantification showing Gata4 expression in the Sham and following injury. Scale bar = 10 μm . (* $p < 0.05$, ** $p < 0.01$, *** $p < 0.001$, **** $p < 0.0001$). Data are shown as mean with SD ($n = 4$ animals per group). **j** Comparative analysis of the most significant pathways across the time points from RNA-seq data. **k** inference of cell type-proportions (RNA-seq deconvolution) obtained using the CellMapper package **l** Comparison analysis between the different time points of the most significant pathways identified using MS data. Two biological replicates deriving from the pooling of 4 different animals ($n = 8$ animals per group) were analysed for RNA-seq and MS experiments.

Fig. S5. Smooth muscle actin expression during zebrafish heart regeneration. **a** Confocal images and **b** relative quantification showing smooth muscle actin (SMA) expression in the Sham and following injury. Cardiomyocytes are identified by GFP positivity. Data are shown as mean with standard deviation ($n = 4$ animals per group). Scale bar = 100 μm . (* $p < 0.05$, ** $p < 0.01$, *** $p < 0.001$, **** $p < 0.0001$).

3. Fig.4 and page 13: Authors claim that the lectin histochemistry mainly labels blood vessels. Based on the low-res pictures shown this seems difficult to claim. it also seems that WGA is labeling some cardiomyocytes in the BZ, which is ignored. Authors should substantiate their claim with an endothelial co-labeling or change the wording to tone down their conclusion about the specific cell type.

Response: We have now provided higher resolution images in which is possible to identify better blood vessels. Moreover, comments regarding specific cell types are not present (page 13, lines 288-292):

*For all these lectins binding was mainly observed in pre-existing and new-forming vessels that can be easily identified based on their shape, indicating the presence of both terminal α -(2,3)- and α -(2,6)-linked sialic acids on these tissue structures (**Figure 4a, c, e**), even though conclusions on the specific cell types cannot be made.*

Fig. 4. Sialic acid distribution in regenerating zebrafish heart identified by lectin histochemistry. a MAA, recognizing α -(2,3)-linked sialic acid, binding in ventricle tissue, and

b MAA binding intensity relative quantification. **c** SNA-I binding to α -(2,6)-linked sialic acid within the ventricle tissue and **d** SNA-I binding intensity relative quantification. **e** WGA binding to sialic acid and GlcNAc residues in the ventricle tissue, and **f** WGA binding intensity relative quantification. The lectin binding quantifications highlighted a similar decrease at 2 dpci and a subsequent increase in the following time points. Cardiomyocytes were identified by GFP positivity. IA: injured area. Data are shown as mean with one SD (n = 4 animals per group). Scale bar = 100 μ m. (*p < 0.05, **p < 0.01, *** p < 0.001, **** p < 0.0001).

4. For all figures: all images of injured fish heart should be shown in a comparable manner in order to make the comparison easier. Images of whole hearts and sections with immunohistochemistry staining should be placed with the apex pointing down and the bulbus up.

Response: We agree with this Reviewer's suggestion and have revised all images displaying whole heart sections with the apex pointing down and the bulbus arteriosus up to facilitate comparison (Fig.4, Fig. S4 and S5 previously displayed)